

# Field comparison of dry deposition samplers for collection of atmospheric mineral dust: results from single-particle characterization

**Andebo Waza[1], Kilian Schneiders[1], Jan May[2], Sergio Rodríguez[3, 4], Bernd Epple[2], Konrad Kandler[1]**

1Atmospheric Aerosol, Institute for Applied Geosciences, Technische Universität Darmstadt, D-64287 Darmstadt, Germany

2Institute for Energy Systems & Technology, Technische Universität Darmstadt, D-64287 Darmstadt, Germany

3Izaña Atmospheric Research Centre, AEMET, Tenerife, Spain.

4Estación Experimental de Zonas Aridas, EEZA CSIC, Almería, Spain.

*correspondence to andebo.waza@geo.tu-darmstadt.de

**Abstract**

Frequently, passive dry deposition collectors are used to sample atmospheric dust deposition. However, there exists a multitude of different instruments with different, usually not well-characterized sampling efficiencies. As result, the acquired data might be considerably biased with respect to their size representativity, and as consequence, also composition. In this study, individual particle analysis by automated scanning electron microscopy coupled with energy-dispersive X-ray was used to characterize different, commonly used passive samplers with respect to their size-resolved deposition flux and concentration. This study focuses on the microphysical properties. In addition, computational fluid dynamics modeling was used in parallel to achieve deposition velocities from a theoretical point of view.

Flux measurements made using different passive samplers show a disagreement between the samplers. Both MWAC and BSNE collect considerably more material than Flat plate and the Sigma-2. The collection efficiency of MWAC for large particles increases in comparison to Sigma-2 slightly with increasing wind speed, while there is barely such increase visible for the BSNE. A correlation analysis between dust flux, derived dust concentrations and wind speed reveals a positive correlation between dust flux and dust concentration and negative correlation between dust flux and wind speed. A very good correlation is found between derived concentrations and $PM_{10}$ concentration measurements by an optical particle spectrometer. The results also suggest that a Big Spring Number Eight as horizontal flux sampler and a Sigma-2 as vertical flux sampler can be good options for $PM_{10}$ measurement, whereas a Modified Wilson and Cooke sample is not a suitable choice. Furthermore, it is found that deposition velocities calculated from classical deposition models do not agree with deposition velocities estimated using computational fluid dynamics simulations. The deposition velocity from CFD was often higher than the



values derived from classical deposition velocity models. Comparatively, deposition velocity calculated
using analytical approach better fits to the measurement data than deposition velocity from CFD.
**Key words**: Mineral dust particles, passive samplers, SEM-EDX, single particle analysis, computational
fluid dynamics

## 1    Introduction

Mineral dust aerosol in the climate system has received considerable scientific attention mainly due to
its direct effect on the radiative budget and indirect one on cloud microphysical properties (Arimoto,
2001; Jickells et al., 2005). Mineral dust particles also play a key part with respect to gas phase chemistry
by providing a reaction surface e.g. ozone depletion (Nicolas et al., 2009; Prospero et al., 1995).
Moreover, dust aerosol also plays an important role in biogeochemical cycles by supplying important
and limiting nutrients to Ocean surfaces. Mineral dust is emitted mainly from the arid and semi-arid
regions of the world and believed to have a global source strength is ranging from 1000-3000Tgyr$^{-1}$
(Andreae, 1995). They form the single largest component of global atmospheric aerosol mass budget,
contributing about one third of the total natural aerosol mass annually (Penner et al., 2001).
Deposition measurement data of mineral dust are useful to validate numerical simulation models and to
improve our understanding of deposition processes. However, the scarcity and the limited
representatively of the deposition measurement data for validation pose a major challenge to assess dust
deposition at regional and global scales (Schulz et al., 2012; WMO, 2011). This is in part linked to the
uncertainties evolving from the use of different and non-standardized measurement techniques.
Commonly, deposition is measured by passive techniques, which provide an acceptor area for the
depositing atmospheric particles. The advantage of these passive samplers is that they operate passively,
resulting in simple and thus cheaper instruments, so that many locations can be sampled at a reasonable
cost (Goossens and Buck, 2012). Moreover, the usual lack of a power supply allows also for unattended
remote setups. However, the most important disadvantage is that collection efficiency and deposition
velocity is determined by the environmental conditions not under control (and frequently also unknown).
That implies on addition, that the sampler shape can have a strong and variable impact of the collection
properties. Also, they may need long sampling time necessary to collect sufficient particles.
While there are papers describing and modeling single samplers (Einstein et al., 2012; Wagner and Leith,
2001a, b; Yamamoto et al., 2006) and a few comparison studies (Goossens and Buck, 2012; Mendez et
al., 2016), nearly previous studies only compare on total mass, thereby neglecting size dependence and





potential comparison biases. Mendez et al. (2016) showed that efficiency of BSNE and MWAC samplers
for collecting PM10 varies with wind speed. Furthermore, Goossens and Buck (2012) found that for
$PM_{10}$, concentrations obtained from BSNE and DustTrak samplers have comparable values for wind
speed in between 2–7 m/s.
The purpose of this study is to assess the particle collection properties of different deposition and other
passive samplers based on single particle measurements and their agreement with theory. From the
available data, also relations of the collected particle microphysics and composition homogeneity
between the samplers will be presented, which can be used as estimators for the comparability of previous
literature data based on the different techniques. To the best of our knowledge, this is the first study to
analyze dry deposition measurements collected using passive samplers by means of a single-particle
SEM-EDX Analysis approach (particularly in the size fraction larger than 10 μm).

## 72    2    Material and methods

### 73    2.1    Sampling location and time

Sahara and Sahel provide large quantities of soil dust, resulting in a westward flow of mineral dust
particles over the North Atlantic Ocean accounting for up to 50% of global dust budget (Goudie and
Middleton, 2001). Owing to proximity to the African continent, the Canary Islands are influenced by dust
particles transported from Sahara and Sahel regions. Therefore, Tenerife is one of the best locations to
study relevant dust aerosol in a natural environment.
For this study, we conducted a two month (July to August 2017) aerosol collection and dry deposition
sampling campaign at Izaña Global Atmospheric Watch observatory (Bergamaschi et al., 2000;
Rodríguez et al., 2015) (28.3085ºN, 16.4995ºW). Sampling was performed on top of a measurement
installation, approximately 2m above the ground (including the inlet heights of the samplers). The
installation was made on a 160m$^2$ flat concrete platform. The trade wind inversion, which is a typical
meteorological feature of the station shields most of the time the observatory from local island emissions
(García et al., 2016). Therefore, Izaña Global Atmospheric Watch observatory is an ideal choice for in-
situ measurements under "free troposphere" conditions (Bergamaschi et al., 2000; García et al., 2016).

### 87    2.2    Wind measurements

An ultra-sonic anemometer (the young models 81000 was installed at approximately 2m height above
the ground to obtain the 3-D wind velocity and direction and was operated with a time resolution of 10
Hz to get basic information on turbulence structure.



### 2.3 Particle sampling

Samples were collected from different, commonly used samplers, namely Big Spring Number Eight (BSNE) (Fryrear, 1986), Modified Wilson and Cooke (MWAC) (Wilson and Cook, 1980), Sigma-2 (VDI2119, 2013) and Flat-plate (UNC-derived)(Ott and Peters, 2008). In addition, the free-wing impactor (FWI) (Kandler et al., 2018) was also used to collect coarser particles. The BSNE, MWAC, FWI and Filter Sampler were mounted on wind vane to align to ambient wind direction. Samples were collected at interval of 24 hours (exposure time). The sampling duration for FWI (12mm Al-stub) was 30min. The sampling duration for filter sampler was set to be one hour.

#### 2.3.1 Flat plate sampler

The flat plate sampler used in this work was taken from the original flat plate geometry used in Ott et al. (2008b). Briefly, the geometry contains two round brass plates (top plate diameter 203 mm, bottom plate 127 mm, thickness 1 mm each) mounted in a distance of 16mm. Unlike the original design, the geometry of the current work has a cylindrical dip in the lower plate, which recedes the sampling substrate – a SEM stub with a thickness of 3.2 mm – from the airflow, reducing the flow disturbance. A preliminary study with identical setup in a rural environment had shown that this recession approximately doubles the collection efficiency for large particles. In this design, larger droplets (> 1 mm) are prevented by this setup from reaching the SEM stub surface at the local wind speeds (Ott et al., 2008b). As described in (Wagner and Leith, 2001a, b), the main triggers for particle deposition on the substrates for this sampler are diffusion, gravity settling, and turbulent inertial forces, of which only the latter two are relevant in our study.

#### 2.3.2 Sigma-2 sampler

The sigma-2 sampling device is described in (Dietze et al., 2006; Schultz, 1989; VDI2119, 2013). Briefly, the geometry consists of a cylindrical sedimentation tube with a height of about 27cm made of antistatic plastic, which is topped by a protective cap with diameter of 158mm. At its top, the cap has four rectangular inlet windows (measuring 40mm x 77mm, all at the same height) at its side providing away for passive entrance of particles to the collection surface. Once entered the tube, particles settle down to the collection surface due to gravitation (Stokes' law) (VDI2119, 2013) . The samplers designed in a way that it prevents the sample from direct radiation, wind and precipitation.



### 2.3.3 The Modified Wilson and Cooke (MWAC) sampler

119

The MWAC sampler is based on an original design developed by Wilson and Cook (1980). The sampler consists of a closed polyethylene bottle, serving as settling chamber, to which an inlet tube and an outlet tube have been added. The MWAC sampling bottles are 95mm long with a diameter of 48mm. The two inlet and outlet plastic tubes with inner and outer diameter 8 and 10mm respectively, pass air through the cap into the bottle and then out again. The large volume of the bottle relative to the inlet diameter makes the dust particles entering the bottle to be deposited in the bottle due to the flow deceleration the total bottle area, and due to impaction below the exit of the inlet tube. The air then discharges from the bottle via the outlet tube. MWAC is one of the most commonly used samplers (Goossens and Offer, 2000) and has a high sampling efficiency for large particles (Mendez et al., 2016).

### 2.3.4 The Big Spring Number Eight (BSNE) sampler

The BSNE sampler originally designed by Fryrear (1986) is intended to collect airborne dust particles from the horizontal flux (Goossens and Offer, 2000). Briefly, the particle laden air passes through a rectangular inlet (21mm wide and 11mm high, with total area of 231mm$^2$). Once inside the sampler, air speed is reduced by continuous cross section increase (angular walls) and the particles settle out in collection surface. Air discharges through a mesh screen.

### 2.3.5 Free-wing impactor (FWI)

A free rotating wing impactor (Jaenicke and Junge, 1967; Kandler et al., 2018; Kandler et al., 2009) was used to collect particles larger than approximately 5μm. A FWI has a sticky impaction surface attached to a rotating arm that moves through air; particles deposit on the moving plate. The rotating arm is moved at constant speed by a stepper motor, which is fixed on a wind vane, aligning the FWI to wind direction. The particle size cut-off is defined by the impaction parameter, i.e. by rotation speed, wind speed and sample substrate geometry. The details of working principle of FWI can be obtained from Kandler et al. (2018)

### 2.3.6 Filter sampler

A filter sampler with Nucleopore filters (Whatman® Nuclepore™ Track-Etched Membranes diam. 25 mm, pore size 0.4μm, polycarbonate) mounted on a wind vane was used for iso-axial particle collection. An inlet nozzle of 6 mm was used to achieve pseudo-isokinetic conditions. Sample flow (0.75m$^3$hr$^{-1}$) was





measured by a mass flow meter (MASS-STREAM, M+W instruments). The filter sampler was operated
at least two times a day.

### 2.4    Upward-downward flux sampler

Following an approach by Noll and Fang (1989) – assuming that turbulent transport is the main
mechanism for upward flux while turbulent transport and sedimentation are the mechanism of for
downward deposition flux – a sampler with an upward- and a downward-facing substrate in analogy to
the flat plate sampler was designed. Both substrates are aligned to face each other with the air passing in
between.

### 2.5    Ancillary Aerosol Data

Additional information regarding the aerosol particle size distributions has been obtained by using OPC
(GRIMM) instruments available at Izaña Global Atmospheric Watch observatory (Bergamaschi et al.,
2000; Rodríguez et al., 2015). The particle size ranging from 10nm to 496nm was measured with an
SMPS while from 350nm to 20µm was measured with an OPC.

### 2.6    SEM-Analysis

All aerosol samples were (except the filter sampler) collected on pure carbon adhesive substrate (Spectro
Tabs, Plano GmbH, Wetzlar, Germany) mounted on standard SEM aluminum stubs (12 and 25mm).
Individual particle analysis by automated scanning electron microscopy (SEM; FEI ESEM Quanta (400
FEG, FEI, Eindhoven, The Netherlands; operated at 12.5 kV, lateral beam extension 3 nm approx., spatial
resolution 160 nm)) was used to characterize particles for size and composition. A total of 315,000
particles from six samplers was analyzed. 26 samples from BSNE (52882 particles), 23 samples from
MWAC (48650 particles), 23 samples from SIGMA-2 (38506 particles), 18 samples from flat plate
(12mm) (24340 particles), 22 samples from Flat plate (25mm) (20700), 13 samples from Filter (80000)
and 12 samples from FWI-12mm (50000 particles) were analyzed. Each sample was characterized at
randomly selected areas, until a total of 3,000 particles with projected area diameters greater than 1µm
was reached. For particle identification, the backscattered electron image (BSE-image) has been used, as
dust particles contain heavier elements than carbon and therefore appear as detectable bright spots in the
BSE-image.
Chemical information was derived by energy-dispersive X-ray analysis (EDX; Oxford X-Max 120,
Oxford Instruments, Abingdon, United Kingdom). The internal ZAF-correction of the software system –
based on inter-peak background radiation absorption measurements used for correction – was used for
obtaining quantitative results.



### 2.7 Particle size determination


The image analysis integrated into the SEM-EDX software determines the size of particles as a projected
area diameter.
$$\mathbf{d_g} = \sqrt{\frac{4B}{\pi}} \qquad (1)$$
Where $\mathbf{B}$ and $\boldsymbol{d_g}$ are the area covered by the particle on the sample substrate and the projected area
diameter respectively.
Following Ott et al. (2008a), the volumetric shape factor, $\boldsymbol{S_v}$ is determined from the count data as:
$$\boldsymbol{S_v} = \frac{P^2}{4\pi A} \qquad (2)$$
Where P and A are the perimeter and the projected area of the particle respectively.
The volume-equivalent diameter (sphere with the same volume as the irregular shaped particle) is then,
calculated from the projected area diameter via the volumetric shape factor (Ott et al., 2008a) and is
expressed by particle projected area and perimeter as
$$\boldsymbol{d_v} = \frac{4\pi B}{P^2} dg = \frac{1}{P^2}\sqrt{64\pi B^3} \qquad (3)$$
The aerodynamic diameter ($d_a$) is calculated from projected area diameter through the use of a volumetric
shape factor and aerodynamic shape factor (Wagner and Leith, 2001b)
$$d_a = \sqrt{[d_v\,(\rho_p/\rho_0)1/S_d)]} \qquad (4)$$
With $\boldsymbol{S_d}$ the aerodynamic shape factor, $\boldsymbol{\rho_p}$ and $\boldsymbol{\rho_0}$ are particle density and air density respectively. For
this work, a value of $S_d = 1.41$ was used (Davies, 1979). Cunningham's slip correction was neglected in
this study, as all particles considered were super-micron size.

### 2.8 Mass flux calculation


The mass flux (M) is calculated from deposited particle number per area, individual particle size and
density. The particle density was assumed to be equal the bulk material density of the dominating
identified compound for each particle (e.g., Kandler et al. 2007). A window correction (Kandler et al.,
2009) was applied to the particle flux as:
$$\boldsymbol{C_w} = \frac{w_x w_y}{(w_x - d_p)(w_y - d_p)} \qquad (5)$$
Where $w_x$ and $w_y$ are the dimensions of the analysis rectangle.





The mass flux of the samples is then determined as
$$M = \frac{1}{Ati} \sum_k \rho \; d_p^3 C_w(d_p, k)$$     (6)
With A is the total analyzed area, t is the sample collection time and k is index of the particle.
Size distributions for all properties were calculated for the logarithmic-equidistant intervals of 1-2µm, 2-
4µm, 4-8µm, 8-16µm, 16-32µm, and 32-64µm.
**2.9     Modeling atmospheric mass concentrations and its size distribution from flux measurements**
Concentrations are calculated from the deposition flux using different deposition velocity models for
different samples, namely the models of Stokes and Piskunov. The basic relationship between
concentration and deposition rate was already given by Junge (1963), as the ratio of deposition flux to
concentration:
$$V_d = F/C$$     (7)
With $F$ is deposition flux and $C$ is concentration.
All different approaches now give different formulations for the deposition velocity, based on a set of
assumptions and neglections.
2.9.1 Stokes settling
Terminal settling velocity ($V_{ts}$) is calculated according to Stokes' law.
$$V_{ts} = \frac{d_p^2 g(\rho_p - \rho_a)}{18\mu}$$     (8)
Where $d_p$ is the particle size, g is the gravitational acceleration (9.81ms[-2]); $\rho_p$ the density of particle; $\rho_a$
the air density; µ is the dynamic viscosity of air (1.8e-05kgm[-1]s[-1]).
2.9.2 Turbulent deposition and more complex deposition models
To calculate the turbulent impaction velocity, which depends of the wind speed, the friction velocity is
needed. Friction velocity (u$_s$), which is a measure of wind generated turbulence is one most important
variables affecting deposition velocity (Arya, 1977). Mainly two different approaches have been used to
estimate u-s. On one hand the momentum flux or the eddy covariance (EC) approach (Ettling, 1996),
which directly estimates friction velocity from the correlations between the measured horizontal and
vertical wind velocity fluctuation, and on the other the law of the wall (LoW) approach (Shao et al.,
2011), which estimates u-s from the wind profile. The latter can be approximated from free-stream
velocity and roughness assumptions (Wood (1981)), where the flow inside the sampler is assumed to be





in the hydraulically smooth regime (Schlichting, 1968). **Figure 1** shows correlations between u-s
estimated using Wood (1981) and Ettling (1996) approaches. Obviously, the approaches lead to different
results, for which no clear explanation is available (Dupont et al., 2018) .

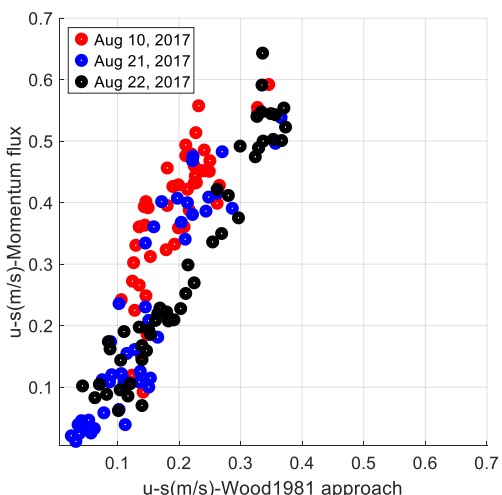


**Figure 1**: Comparison of the friction velocities obtained from the momentum flux and the Wood1981
approaches for different days with different wind speeds (average wind speed =2.900m/s, 2.075m/s,
3.110m/s for Aug 10, 2017, Aug 21, 2017, Aug 22, 2017 respectively).
For the current work, the friction velocity is calculation is based on Wood (1981) approach:
$u\_s = (u/\sqrt{2}) \left[ (2 log10(Re) - 0.65)^{-1.15} \right]$                    **(9)**
Where Re is the flow Reynolds number at the sampling stub location and is given as
$Re = uX/V$                                            **(10)**
$X$ is the distance from the lower plate edge to the center of the sampling stub (6.3cm) and $V$ is kinematic
viscosity.
The reason why we opted to use the Wood (1981) over the Ettling (1996) approach is a) its simplicity, as
it requires only average wind speeds instead of 3D high resolution ones, and therefore will be more
commonly applicable; and b) the fact that the momentum approach yields sometimes uninterpretable
data, in particular in case of buoyancy-driven flow.
There are a variety of models estimating the particles deposition speed (Aluko and Noll, 2006; Noll and
Fang, 1989; Noll et al., 2001; Piskunov, 2009; Slinn and Slinn, 1980; Wagner and Leith, 2001a) (see



**Figure 2**). The formalism of Piskunov (2009) deposition speed model was selected for calculation of concentration in this work. Unless otherwise stated, the particle density used in deposition velocity calculation is 2600kgm$^{-3}$.

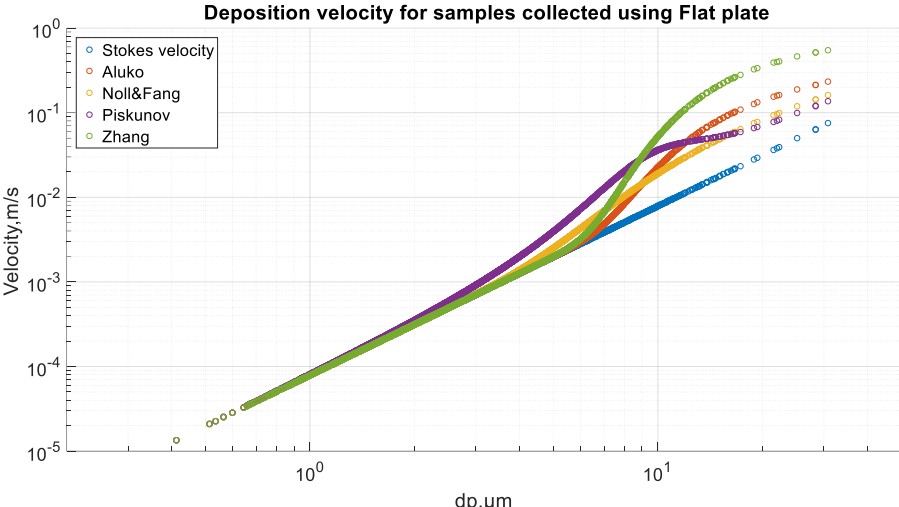

**Figure 2**: Deposition velocities for single particles to a smooth surface calculated by using set of different classical deposition models for Tenerife samples (Aug 9, 2017; average wind speed =3.045m/s).

2.9.3 Deposition models applied to the samplers

The Piskunov deposition velocity model was used for flat plate and BSNE samplers, as in both a horizontal flow deposits particles onto a horizontal flat substrate. , For the sigma-2 sampler, it is assumed that each particle settles with the terminal settling velocity (Tian et al., 2017) and therefore, Stokes' velocity was used for calculation of mass concentrations. In the case of MWAC, a different approach was required due to its semi-impaction geometry. We derived a velocity model based on wind speed (or a reduced wind speed) and calculated the collection efficiency assuming the MWAC to act as impactor for particles in the range of the cut-off diameter and larger. For smaller particles, we assumed that flow is like a flow over a smooth surface, so the Piskunov deposition velocity model was applied. I.e. as soon as the deposition velocity from impactor considerations becomes smaller than the Piskunov one, the latter was used.





### 2.10 Determining the size distributions for mass concentration from the free-wing impactor measurements

Considering the windows correction and the collection efficiency dependence on the impaction speed and geometry, the overall collection efficiency is calculated according to Kandler et al. (2018).

After calculating the collection efficiency, the atmospheric concentration is calculated from flux and deposition velocity as

$$C = \frac{M}{V_d} \tag{11}$$

With Vd = E*v_imp, E being the collection efficiency and v_imp the impaction velocity, calculated from ambient wind speed and rotation speed.

### 2.11 Determining the size distributions for mass concentration from the filter sampler measurements

Apparent number concentrations are determined from the particle deposition rate and the volumetric flow rate calculated from the mass flow for ambient conditions. The inlet efficieny ($\boldsymbol{Eff_{in}}$) – accounting for the difference in wind speed and inlet velocity - is calculated as a function of Stokes number (Stk). The ambient concentration finally ($\boldsymbol{N_{out}}$) is calculated by weighting the measured number concentration with the calculated inlet efficiency correction.

### 2.12 Statistical uncertainty

Owing to the discrete nature of the particle size measurement, the uncertainty coming from counting can pose a significant contribution to the uncertainty of mass flux measurement (Kandler et al., 2018). It is, therefore, important to assess the uncertainties in our mass flux measurements, which is done in accordance to the previous work (Kandler et al. 2018). For the mass flux calculations, the statistical uncertainty is assessed by a bootstrap simulation approach using Monte Carlo approximation (Efron, 1979).

In this work, the bootstrap simulations and the two-sided 95 % confidence intervals calculation were performed by using Matlab's bootstrap function (MATLAB R2016a (MathWorks,Inc). Here, MATLAB function uses a non-parametric bootstrap algorithm (Neto, 2015) to compute the 95% bootstrap confidence interval.



## 3 Computational fluid dynamics (CFD) simulation

Computational fluid dynamics (CFD) simulations were conducted to predict the deposition of particles on to different passive samplers (MWAC, Sigma-2 and Flat-plate). A discrete phase model without interaction with continuous phase was used to calculate the trajectories of the particles. The CFD software ANSYS-FLUENT 18.2 was used for performing the numerical simulations.

### 3.1 Evaluating the mean flow field

In a first step the geometry of samplers was created using ANSYS DesignModeler. In a second step, an enclosure around the geometry was generated. To ensure that there are no large gradients normal to the boundaries at the domain boundary, the domain was created depending on the width, the height and the length of the geometries. The space in front of the geometry is two times the height of the sampler, the space behind the sampler is ten times the height, the space left and right of the geometry is five times the width of the geometry and the space below and above the sampler is five times the height.

Afterwards a mesh was created using the ANSYS Meshing program. For the enhanced wall treatment the first near-wall node should be placed at the dimensionless wall distance of $y^{+\wedge} \approx 1$. The dimensionless wall distance is given by

$$y^+ = \frac{u_* y}{v} \tag{12}$$

With $y$ the distance to the wall, $v$ the kinematic viscosity of the fluid and $u_*$ the friction velocity which is defined for this purpose by

$$u_* = \sqrt{\tau_w / \rho} \tag{13}$$

With $\tau_w$ the wall, shear stress and $\rho$ the fluid density at the wall. The wall is then subdivided into a viscosity-affected region and a fully turbulent region depending on the turbulent Reynolds number $Re_y$

$$Re_y = \frac{\rho y \sqrt{k}}{\mu} \tag{14}$$

With $y$ the wall-normal distance from the wall to the cell centers, $k$ the turbulence kinetic energy and $\mu$ the dynamic viscosity of the fluid. If $Re_y > 200$ the k-epsilon model is used. $Re_y < 200$ the one-equation of Wolfstein is employed (Chmielewski and Gieras, 2013; Fluent, 2015). The flow field was calculated by solving the Reynolds Averaged Navier Stokes's equations with the software ANSYS Fluent. Standard k-epsilon model was used to calculate the Reynolds-stresses. The boundary conditions at the sides of the domain were set to symmetric. The inlet boundary condition was set to 2, 4 or 8 m/s with air as fluid





(Density: 1.225kgm$^{-3}$, viscosity: 1.7849e-05kgm$^{-1}$s$^{-1}$). The outlet boundary condition was set to pressure
outlet.
In the last step, particles were injected into the velocity field. Different particle sizes (1, 2.5, 5, 10, 20
and 50 µm, Stokes' diameter) for three different wind speeds (2, 4, 8 m/s) were investigated. The particles
density was set to a value of 2600 kg/m³ to match an approximate dust bulk density. The number of
particles trapped in the deposition area was determined. The deposition velocity $V_d$ was calculated by
$$V_d = \frac{N_{pt}v}{A_d C_p}$$ (15)
with $N_{pt}$ the number of trapped particle at the deposition area, $v$ the velocity of the air at the inlet
boundary of the domain, $A_d$ the deposition area and $C_p$ the particle concentration at the particle injection
area (Sajjadi et al., 2016). The particle concentration was 4*10$^8$ m$^{-2}$ in all cases, while the injection area
was adjusted to the geometries. The areas are shown in **Figure 3** with 10 exemplare particle trajectories
along with the sampler geometry.
The turbulence intensity **T$_i$** was calculated and plotted from
$$\mathbf{T_i} = \frac{\left(\frac{2}{3}\mathbf{k}\right)^{1/2}}{\mathbf{v}}$$ (16)
With k the turbulence intensity and v the velocity at the inlet of the domain.
**3.2    Sampler geometries**
Detail of the sampler construction are found in the electronic supplement (see **Figure S 4, Figure S 5,**
**Figure S 6).**
3.2.1    Flat plate sampler
Two different cases were calculated for the flat plate sampler (**Figure 3**), a deposition area diameter of
12 mm and another of 25 mm.








**Figure 3**: Geometries of Flat plate sampler (top), Sigma-2 sampler (middle), MWAC sampler (bottom)

CFD modeling domain and velocity magnitude, inlet velocity: 4m/s (right); in addition, the injection

area is shown in black (Flat plate sampler: width 0.2 m, height 0.05 m; Sigma-2-sampler: width 0.2 m,

height 0.1 m; Bottle sampler: width 0.1 m, height 0.05 m) along with exemplary streamtraces.





A mesh with 3920000 cells was generated and the flow field was calculated. In **Figure 3**, the velocity
magnitude in the middle of the domain is shown for a velocity of 4 m/s at the inlet. 4000000 particles
were injected. The deposition area boundary condition for DPM was set to "trap" and the walls were
defined as reflecting boundaries.
3.2.2    Sigma-2 sampler
The geometry of the Sigma-2 sampler is given in **Figure 3**. A mesh with 7600000 cells was generated
and the flow field was calculated. All wall boundary conditions were set to "trap" for the DPM model.
3.2.3    MWAC sampler
In **Figure 3** the geometry of the MWAC sampler is shown. A mesh with 4620000 cells was generated
and the flow field was calculated. All wall boundary conditions were set to "trap" for the DPM model.
**3.3    Velocity contours and vectors for the samplers**
3.3.1    Flat Plate Sampler
The results in the cross section of the domain are shown in **Figure 4.** The formation of the boundary layer
at the wall of the sampler is clearly visible at all velocities. At the central sampling location, the flow
between the plates has the same velocity as the free stream, so for the analytical deposition models, the
lower plate can be treated as single surface. The highest velocity is found at the sharp edge at the bottom
of the sampler. Due to the high velocity gradients in this part there is also the highest turbulence intensity
in the domain. As expected, the turbulent wake becomes smaller with increasing wind speed.






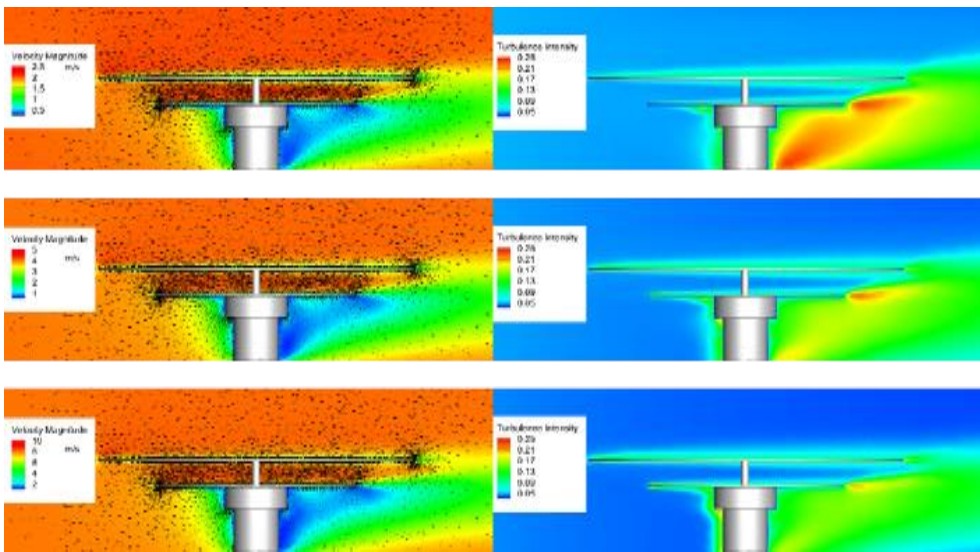


**Figure 4**: Flat Plate Sampler: Velocity magnitude and turbulence intensity at wind speed 2 m/s (top),

Flat Plate Sampler: Velocity magnitude and turbulence intensity at wind speed 4 m/s (middle), Flat

Plate Sampler: Velocity magnitude and turbulence intensity at wind speed 8 m/s (bottom).

3.3.2   Sigma 2 Sampler

The results in the cross section of the domain are shown for the 4 m/s case in **Figure 5.** Apparently the
velocity magnitude inside the sampler is much smaller than outside. In the vertical settling tube, the
turbulence intensity is low, justifying the idea of Stokes settling inside. Owing to the open, but bulky
geometry, there is a flow into the interior at the back. The highest velocities and turbulence intensities
are found at the sharp edges at the top and bottom of the sampler.

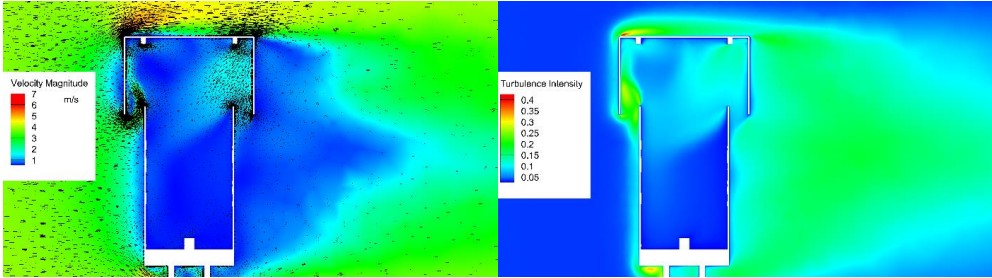

385

**Figure 5**: Sigma-2 Sampler: Velocity magnitude and turbulence intensity at wind speed 4 m/s.



### 3.3.3   MWAC Sampler

The results in the cross section of the domain are shown for the 4 m/s case in **Figure 6**. Furthermore, the velocity field and the velocity vectors in the cross sections across and along the inlet tube are shown in **Figure 7**. In the tubes the typical pipe flow is formed. In the figures showing the cross sections along the inlet tube a symmetrical flow over the pipe cross section is visible.

In **Figure 8** the mean flow velocity in the MWAC tube is shown as a function of the outside velocity for the three cases. The fitting curve shows that the mean velocity in the pipe increases linearly with the external velocity.

In comparison to the other geometries, the turbulent wake related to the geometry size is much bigger for the MWAC sampler.

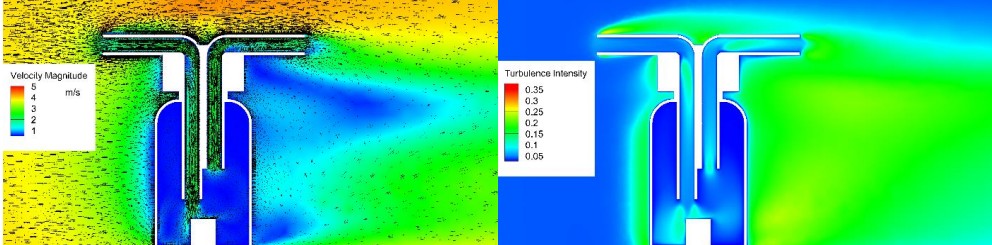

**Figure 6:** MWAC Sampler: Velocity magnitude and turbulence intensity at wind speed 4 m/s.





**Figure 7**: Velocity vectors at 2, 4 and 8 m/s (cross sections across and along the inlet tube).

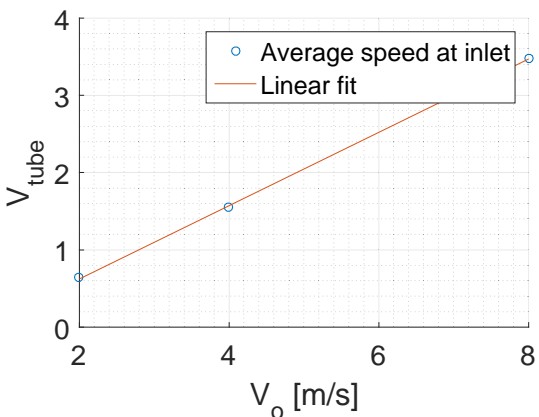


**Figure 8:** Mean flow velocity ($V_{tube}$) in the MWAC tube as a function of the outside velocity ($V_O$).
Fitting curve: $V_{tube} = 0.47V_0 - 0.33$ for the range $2 - 8$ m/s

















## 4 Results and Discussion

### 4.1 Methodical aspects (Field Measurements)

4.1.1 Mass flux comparison between the samplers

Mineral dust was found to be the dominating particle type during this campaign, consisting of different silicates, quartz, calcite, dolomite, gypsum as reported previously for this location (e.g (Kandler et al., 2007)). Therefore, hygroscopicity was not taken into account, as due to the mostly non-hygroscopic compounds and the moderate humidities their impact was rated low. Details on the composition will be reported in a companion paper.

The mass and number fluxes (given per unit time and sample surface area) along with daily average temperature and wind speed are presented as daily values. Details for all days and all samplers can be found in the electronic supplement (**see Table S1, S2, S3, and S4 in the electronic supplement**).

**Figure 9** shows as example mass fluxes for different samplers during a dust event and a non-dust event day. For all samplers, the mass flux size distributions peaked in the 16-32µm interval. This result is in support of the conclusion that atmospheric dry deposition is dominated by coarse particles owing to their high deposition velocities (Davidson et al., 1985; Holsen et al., 1991). There is a considerable difference among different samplers, in particular for the size range with the highest mass deposition fluxes, whereas the difference is small for smaller particles. MWAC and BSNE – both horizontal flux samplers - collect more coarse material than Flat plate and Sigma-2 samplers, which in contrary measure the vertical flux. In particular, the MWAC sampler collects considerably higher coarse particle mass fluxes, probably owing to its impactor-like design.

**Table 1.** The campaign maximum and minimum fluxes measured by the samplers

| Samplers | Maximum flux (mg/ (m$^2$d)) | Minimum flux (mg/ (m$^2$d)) |
|----------|------------------------------|------------------------------|
| MWAC | 1240 | 0.6 |
| BSNE | 310 | 0.2 |
| Flat plate | 80 | 2.0 |
| Sigma-2 | 117 | 1.9 |

From **Table 1** it becomes obvious that the vertical flux instruments collect much less than material than the horizontal flux ones, in accordance with previous findings (Goossens, 2008). In the present study, horizontal to vertical flux ratio as function of particle size is approximately in between 2 and 50, while



Goossens (2008) reported it to be in between 50 and 160. This difference in the ratio might come from
the different approaches. Goossens (2008) used water as a deposition surface while in our study we used
a sampling substrate – a SEM stub suited inside a Flat plate geometry as deposition surface. MWAC
sampler is used a horizontal dust flux sampler in both studies. Furthermore, from **Figure 9**, we can clearly
see that that there is high temporal variation in deposition flux between dust event days and non-dust
event days. Generally, the temporal variation is much higher than difference between samplers.

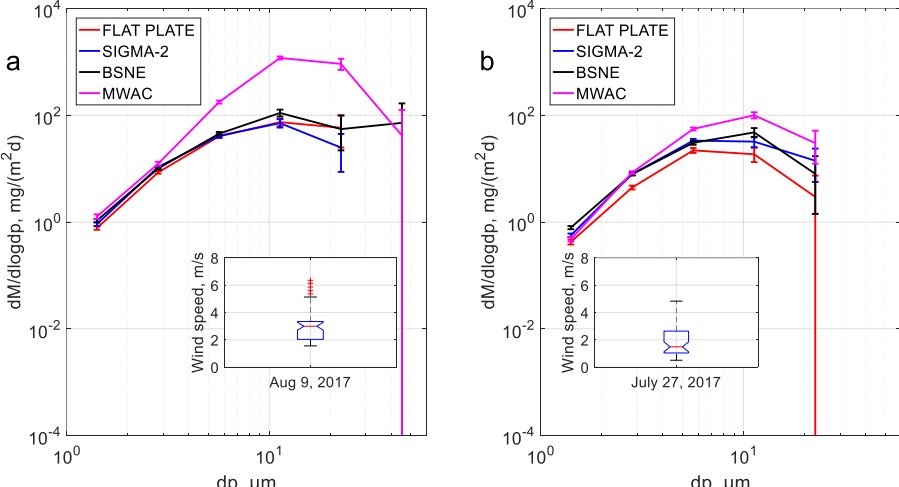

**Figure 9**: Size resolved mass flux measured by different passive samplers: a) dust event day; b) non-
dust event day. Error bars show bootstrapped 95% confidence interval. The inserts show box plots for
the wind speed distribution based on 30-min intervals.
**4.1.1.1  Comparison  in terms as function of wind speed and particle size**
The daily box-plots of 30-min averaged wind speed at Izaña is shown in **Figure 13**. The average wind
speed during the campaign was about 3.5 m/s with the lowest daily median around 1.5 m/s and the highest
7 m/s.
**Figure 10** show the mass flux ratio of MWAC, BSNE and Flat plate to Sigma-2 as function of wind
speed. The collection efficiency of MWAC for large particles increases in comparison to Sigma-2 slightly
with increasing wind speed, while there is barely such increase visible for the BSNE. Both – being
horizontal sampler – collect considerably more material than the Sigma-2.





Similarly, **Figure 11** shows the deposition flux ratio of MWAC, BSNE and Flat plate to Sigma-2 against
particle size. On average, the mass flux ratio of Flat plate to Sigma-2 against wind speed and particle size
is less than one. This indicates that, on average at a given wind speed and particle size, Sigma-2 sampler
collects more particles than flat plate.

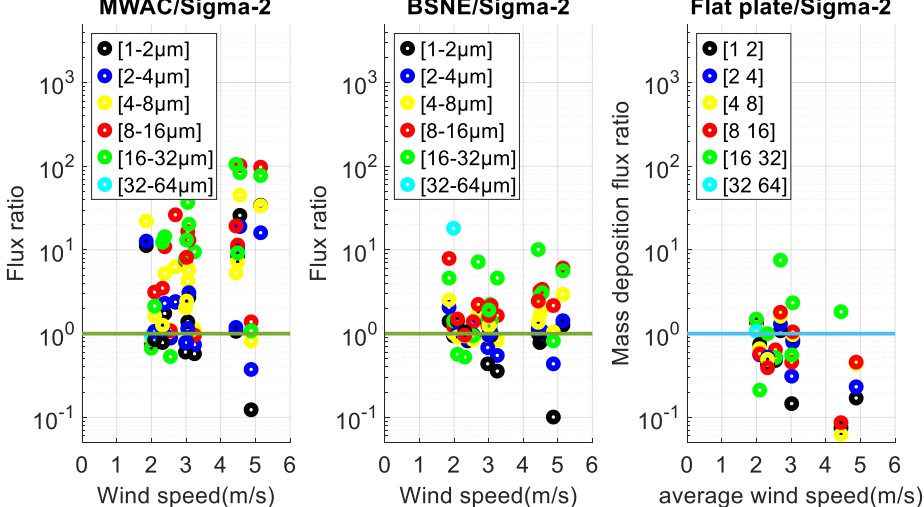


**Figure 10:** Flux ratio as function of wind speed for different days (MWAC/SIGMA-2 and
BSNE/SIGMA-2). Different colors represent different size intervals.

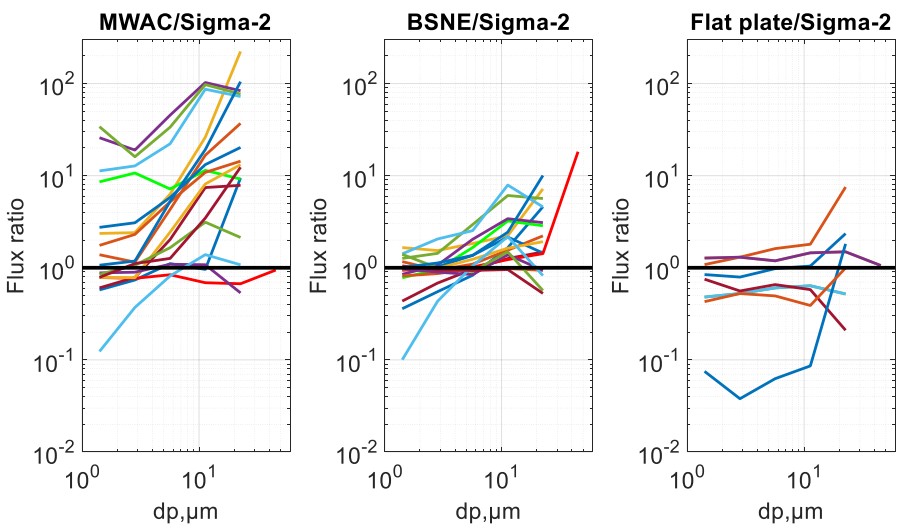





**Figure 11**: Flux ratio as function of particle size (MWAC/Sigma-2, BSNE/Sigma-2 and Flat plate/ Sigma-2). Different colors represent different measurement days.

In connection to this, the ratios of dry deposition flux (number) and deposition velocity from models are shown in the **Figure 12**. The deposition velocity ratio from models is often higher than the ratios derived from the mass and number (greater than factor of 2) due to reasons that are yet to be fully understood and clarified.

The deposition velocity ratio from models also shows in increase with particle size, but up to some size limit and then starts decreasing. The increase in flux ratio with particle size confirms that particle size, among others, strongly affects the deposition velocity for particles.

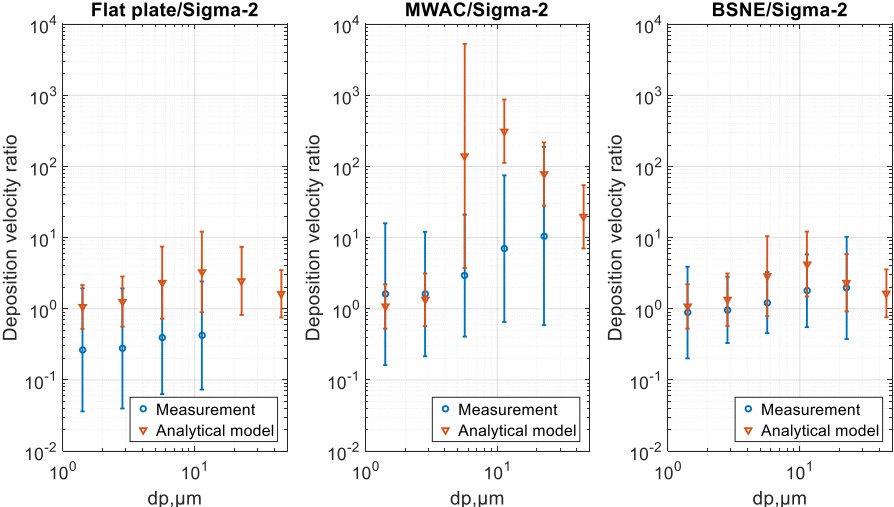

**Figure 12:** Comparing geometric mean ratio of fluxes (Flat plate/Sigma-2 , MWAC/Sigma-2 , BSNE/Sigma-2 ) of measurements to geometric mean ratio of deposition velocity calculated using models (Flat plate; Piskunov, MWAC; Piskunov & Sigma-2; Stokes). Error bars show geometric standard deviations.

4.1.2   Dependence of small particle dust deposition on atmospheric PM$_{10}$ concentration and wind speed

**Figure S 2** (in the electronic supplment) and  **Table 2** display the correlation between deposition number fluxes, atmospheric number concentration by the OPC and the wind speed for different samples. It is evident (**Figure S 2)** that there is generally a positive correlation between concentration and number flux, which can be expected from the high variation in concentration compared to the lower variation in wind





speed. However, deposition rate and wind speed are even anti-correlated, indicating a cross-influence of wind speed and concentration. Such a behavior was observed by different techniques for a dust transport region (Kandler et al., 2011). Also, an ambiguous wind-dependency has been reported for other places (Xu et al., 2016). The main driver of the deposition rate during this study is obviously the dust concentration.

**Table 2**: Summary of regression analysis for correlation between dust deposition flux vs atmopsheric concentration and deposition flux vs wind speed. Significant relationships are shown in bold.

|  | Flux vs concentration | | | Flux vs wind speed | | |
|---|---|---|---|---|---|---|
|  | $r^2$ | p-value | slope (m/d) | $r^2$ | p-value | Slope ($1.16*10^5$ /($m^3$)) |
| Flat plate | 0.606 | 0.005 | 0.58 | 0.315 | 0.0726 | -0.28 |
| MWAC | 0.287 | 0.171 | 0.21 | 0.391 | 0.0974 | -0.17 |
| BSNE | **0.975** | **$1.5 * 10^{-8}$** | **0.87** | 0.014 | 0.729 | -0.05 |
| Sigma-2 | **0.877** | **0.0002** | **0.78** | 0.0128 | 0.772 | -0.07 |

In a second step the correlation between modeled dust concentration from different samplers and the corresponding OPC-measured concentration was assessed **Table 2.** However, no increase in correlation is observed, indicating that – like already observed from the ratio calculations above – the deposition models fail to describe the deposition behavior in detail.

From the correlation relations it can be learned that MWAC is least suitable for estimating $PM_{10}$, which fully agrees well with previous studies (Mendez et al., 2016). However, the correlation analysis here shows that BSNE is actually a suitable instrument for a $PM_{10}$ estimation, which is in contrast to the wind-tunnel observation of (Mendez et al., 2016). This discrepancy might be derived from the different approaches. While in the referred previous work the loss of concentration from the passing aerosol was measured, here a gain of deposition was investigated. As result, for lower deposition velocities (discussed below), the former approach will yield high uncertainties. Similar to BSNE, flat plate and Sigma-2 appear good estimators for $PM_{10}$ , which is also in accordance with previous studies (Dietze et al., 2006).



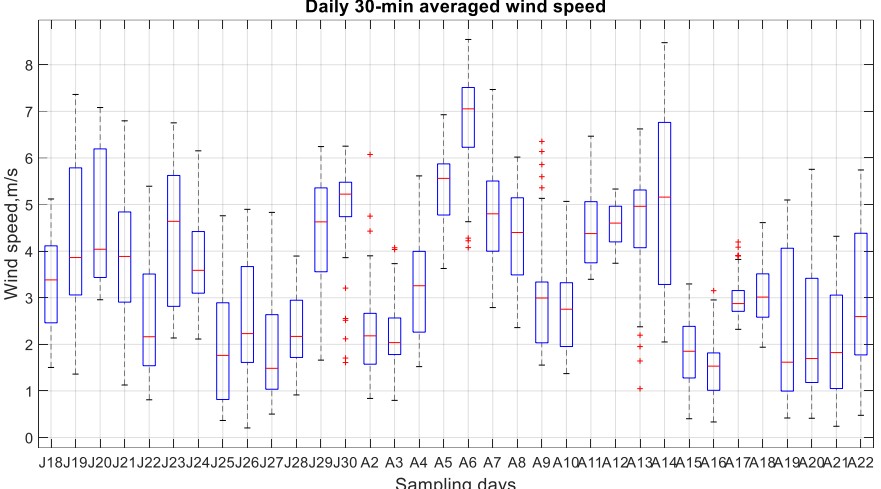



**Figure 13:** Daily box-plots of 30-min averaged wind speed observed at Izaña Atmospheric Observatorio from 18/July/2017 to 23/August/2017. The black vertical lines show the standard deviation (J=July, A=August).

**4.1.2.1 Small particle apparent deposition velocity** (PM$_{10}$ **size range: 1-2µm, 2-4µm, 4-8µm**)
**Figure 14** displays the apparent deposition velocity (the ratio of the number flux to the concentration of
the OPC, for each particle size class) as function of wind speed for different samplers. Obviously, there
is not clear trend for the small particles. The effective deposition velocities range between $3.5*10^{-6}$-
$5.7*10^{-4}$m/s. As can be clearly seen from the plot, the effect of wind speed on deposition velocity is
negligible.



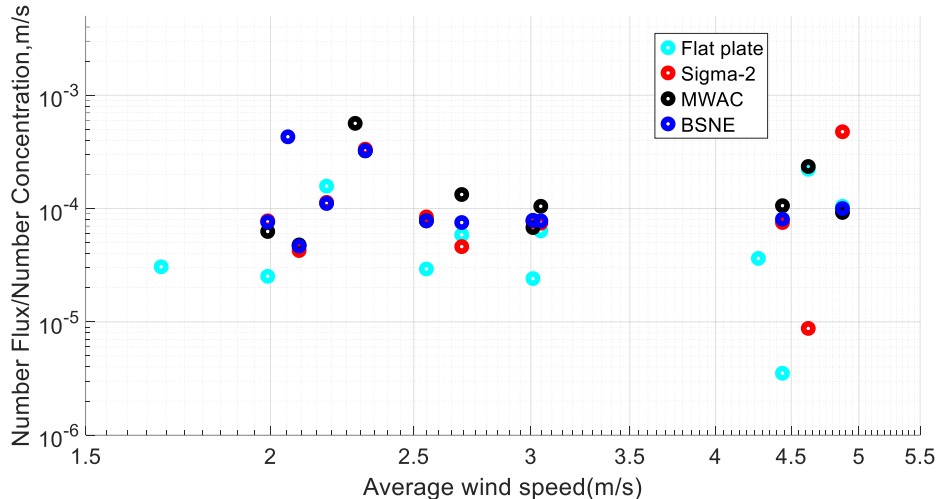


**Figure 14**: Apparent deposition velocity: ratio of number flux (approximately **PM₁₀** size range) to number concentration (OPC; approximately **PM₁₀** size range) as function wind speed.


4.1.3    Atmospheric mass concentration calculation from deposition flux
**4.1.3.1    Consistency between samplers**
**Figure 15** compares a mass flux size distribution with the according concentrations derived by the
modeled deposition velocities. Mass concentrations calculated from different passive samplers agree
generally well with respect to the statistical uncertainties, which is the case for most of the days (**see also**
**Figure S1 in the electronic supplement**). This indicates that the deposition velocity models selected for
the samplers are generally suitable, despite the deviations in single cases.




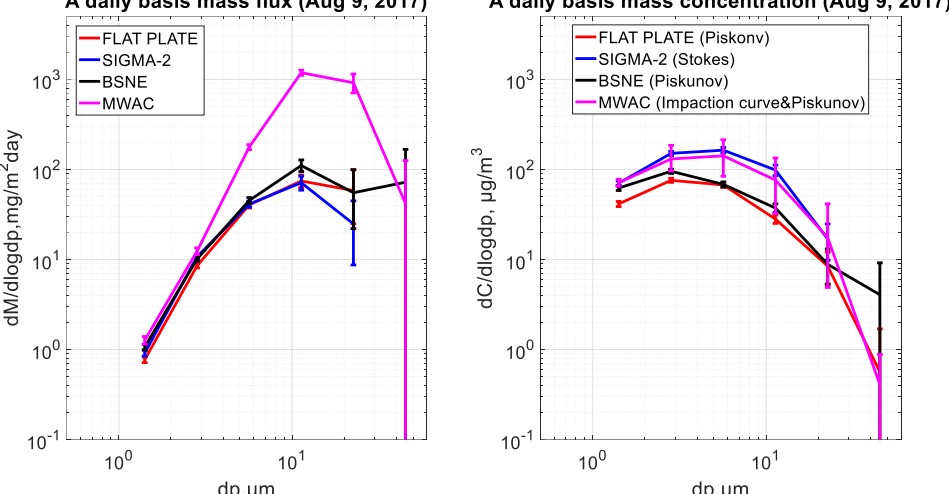


**Figure 15:** Comparing dust mass flux and dust mass concentration. Bars show 95% confidence interval.

### 4.1.3.2 Size-resolved comparison with active samplers

The calculated number concentration in size interval between 1–10μm is validated through a comparison with concentration measured using OPC (Grimm). Similarly, the mass concentration size distribution above PM$_{10}$ size range is validated using the FWI measurements.

**Figure 16** shows the comparisons of number size distributions calculated from flux measurements of the flat plate and MWAC sampler with ones measured using the OPC for different days. Overall number concentrations obtained from OPC measurement are slightly higher than the ones from the fluxes. To the contrary, for low-dust days they are slightly smaller.




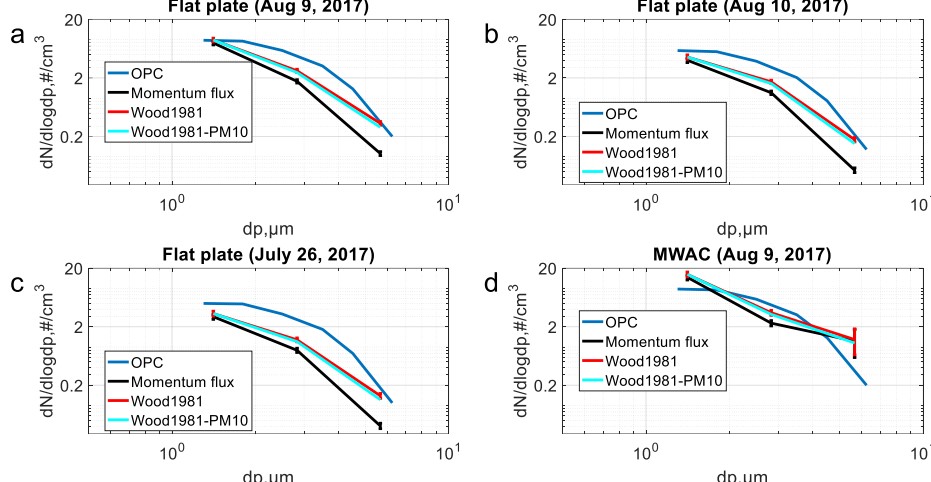

544

**Figure 16:** Comparing number concentration from measurement with number concentration by OPC

measurement (Flat plate sampler (a), (c), (d), and MWAC (b); daily average wind speed=3.05m/s,

2.69m/s, 2.28m/s and 2.55m/s for Aug 9, 2017,  Aug 10, 2017, Aug 2, 2017 and July 2017

respectively). The light blue curve shows the concentration curve calculated using (Piskunov – Wood

approach) with $PM_{10}$ inlet correction (atmospheric concentration), the red curve shows the

concentration curve calculated using (Piskunov – Wood approach);  The green curve shows the

concentration curve calculated using (Piskunov – Momentum flux approach); The dark blue curve

shows the concnetration measurement by OPC. Bars show the central 95 % confidence interval

In general, **Figure 16** show that the deviation of calculated values from OPC measured values is

significant.

In this connection, the above figure (**Figure 16**) also show the comparison of the mass concentration

size distribution measurement obtained by eddy covariance method of u-s estimation (Ettling, 1996)

and the size distribution measurement obtained with friction velocity estimated using Wood (1981)

approach. As shown in the figures, the mass concentration deduced using friction velocity estimated

from Wood (1981) formulation appear larger than the ones deduced from the momentum flux and

therefore agree in our case better with OPC data.

For a comparison with large particles, measurements of FWI are used (**Figure 17).**



562

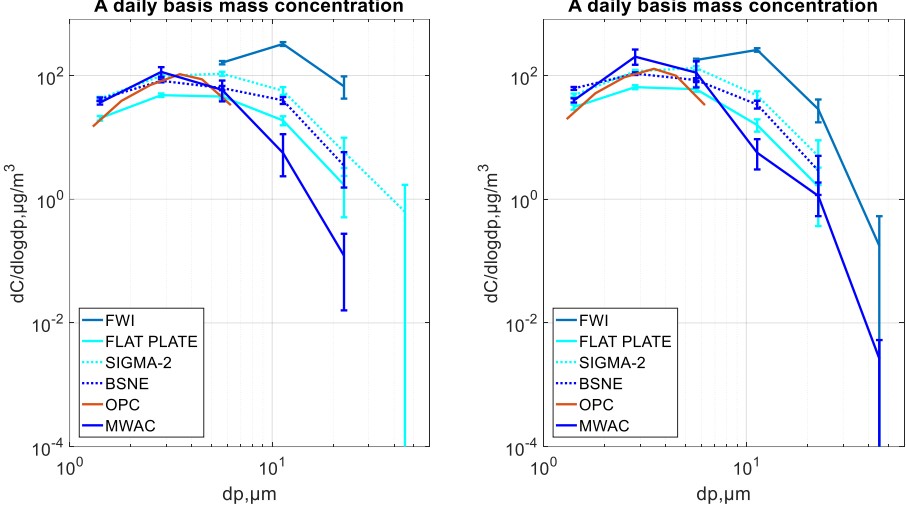

563

**Figure 17:** a daily basis mass concentration measured with passive-sampler method, in comparison to active samplers (FWI). Error bars show the central 95% confidence interval.

Here, a large inconsistency occurs between mass size distribution by passive samplers and by FWI. In particular, the size range larger than 10 µm seems to be largely underestimated by the passive samplers. While for particles around 10 µm, this could be partly to a badly-defined collection efficiency curve of the FWI (Kandler et al. 2018; 50 % cut-off at 11 µm) and the according correction, this can't be the reason for the large particles, where this efficiency approaches unity. Here, either the deposition velocity for the samplers is apparently overestimated.

A further comparison of deposition-derived concentrations with these determined from the iso-axial filter sampler (**Figure 18**) shows that, while the calculated size distributions are in good agreement with the OPC ones, the filter-derived seem to relatively underestimate the concentrations.

Moreover, a correlation analysis (R-squared: 0.947, p-value = 0.0053and slope = 2.0733) suggests that there is a positive correlation between calculated number concentration from filter samples and the OPC measured concentration.





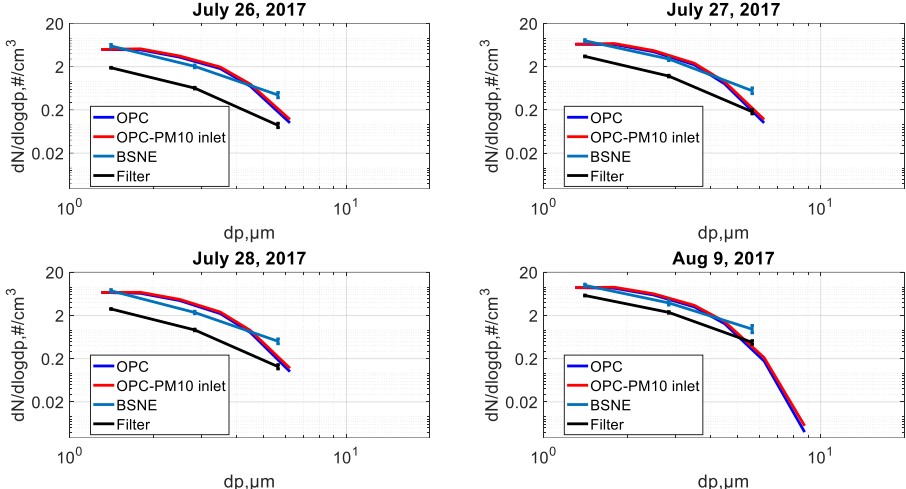


**Figure 18:** Number concentration measured with Filter-sampler method, in comparison to BSNE and OPC. The red curve shows OPC with $PM_{10}$ inlet efficiency correction.

4.1.4    Estimating the turbulent versus gravitational transport fraction by upward-/downward-facing

582          measurements

Details of size resolved mass and number flux measurements along with daily average temperature and
wind speed for up-ward and down-ward flux is given in the electronic supplement (**see table S5 and S6**).
The upward flux is always less than the downward flux. This is expected because the upward facing
substrate (for the downward flux) collects particles deposited by gravitational settling and turbulent
inertial impaction, while the downward facing substrate (for the upward flux) collects particles only by
means of turbulent impaction. **Figure 19** shows the mass and number flux ratio of upward flux to
downward flux as function of particle size. The deviation is greatest for the particle size range around 8
µm, which are strongly affected by turbulence (Noll and Fang, 1989). However, only a very weak trend
of increasing ratio with increasing wind speed can be found here (see **Figure S 3** in the electronic
supplement). Besides the wind speed magnitude, different properties were calculated (e.g, turbulent
intensity, Monin-Obukhov length, relative standard deviation of wind speed, average vertical
component), but none of them was able to explain the observed variations in the flux ratio.





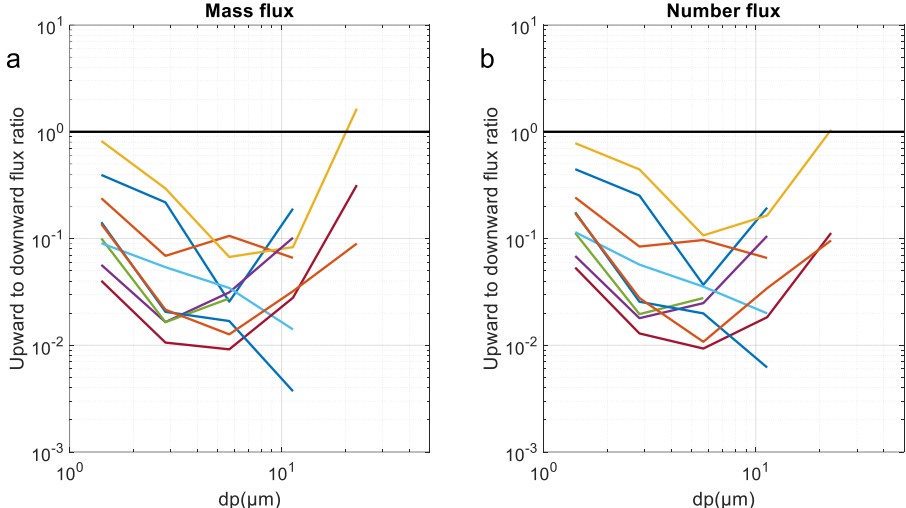


**Figure 19**: Upward to downward flux ratio vs particle size. The flux is measured using flat plate sampler (with 25mm stub), Mass flux (a) and number flux (b). Different colours represent different measurement days.

### 4.2 Computational fluid dynamics (CFD) simulation

Using computational fluid dynamics (CFD), deposition velocities of particles for different passive samplers were predicted and compared to the analytical deposition velocity models used for the different samplers (see **Figure 20**).

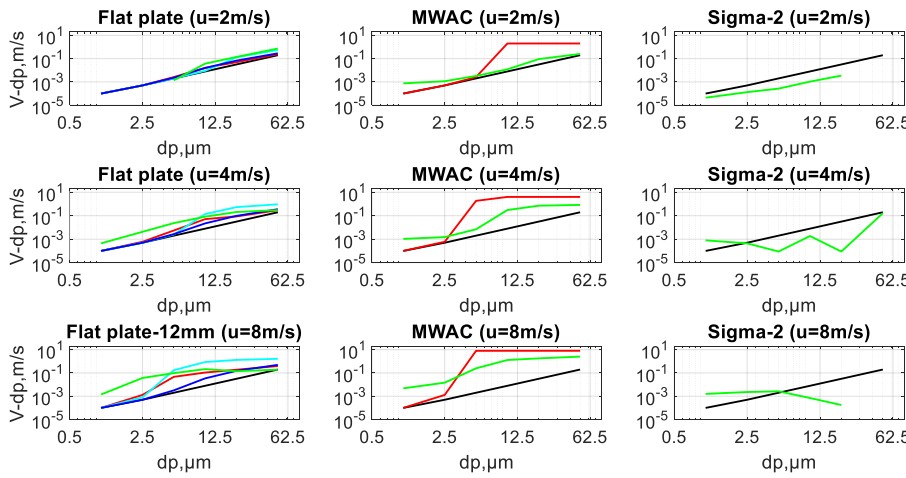




**Figure 20**: The red curve shows deposition velocity calculated using Piskunov deposition velocity
model, the black curve shows deposition velocity calculated using Stokes's velocity, the blue curve
shows deposition velocity calculated using Noll and Fang deposition velocity model, the cyan curve
shows deposition velocity calculated using Zhang deposition velocity model, the green curve shows
deposition velocity from CFD.
While for the flat plate and MWAC sampler the curves agree qualitatively, for the Sigma-2 except for
the lowest wind velocity, they are largely contrary. The latter might be owed to the fact that in a flow
model the non-omnidirectional construction of the Sigma-2 might lead to preferred airflows, which are
not relevant in a more variable and turbulent atmosphere. However, also for the former ones, the
deposition velocity curves are quantitatively different. **Figure S 7** in the electronic supplement shows
comparison of the CFD-derived particle deposition velocities at different wind speed values for different
samplers.

### 4.3    Comparison of measured deposition flux ratios to analytically and CFD modeled ones

**Figure 21** shows comparison of correlations of the deposition velocity ratios derived from the analytical
models (left column) with the according measured deposition velocity (taken as according flux ratio)
with the correlation of the ratios derived from CFD modeling with the measurement. As the CFD models
could only be calculated for a limited number of flow velocities, deposition velocity values were
interpolated between the calculated cases. Generally, the agreement is very low. Practically no variation
observed in the measurement data can be explained by model variation, independent of the type of model.
While this might be explained to a smaller extent by the propagating measurement uncertainties for the
largest particles with low counting statistics, for the smaller ones this systematic deviation must have
other reasons.



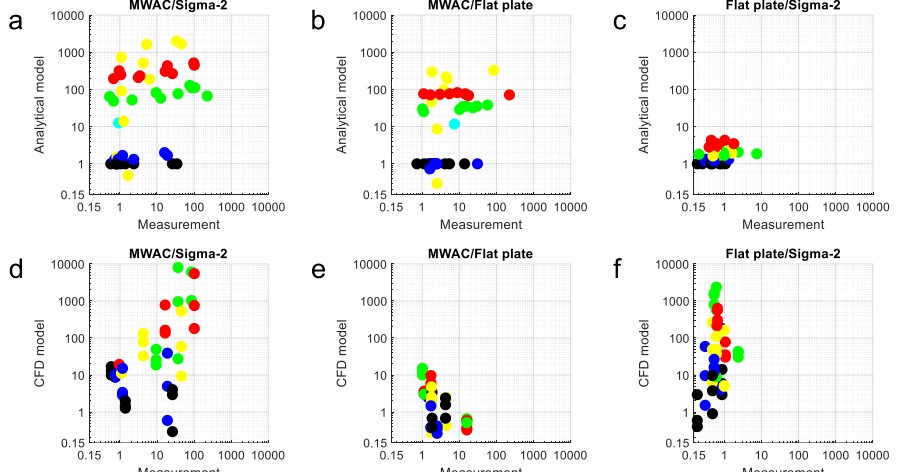

**Figure 21:** Comparison of deposition velocity ratios (MWAC/Sigma-2; measurement to analytical

model (a); measurements to model, MWAC/Flat plate; measurement to analytical model (b), Flat

plate/Sigma-2; measurement to analytical model (c), MWAC/Sigma-2; measurement to CFD model

(d), MWAC/Flat plate; measurement to CFD model (e), Flat plate/Sigma-2; measurement to CFD

model (f) plate/Sigma-2). Different colors represent different size intervals of different measurement

days; 1-2µm: Black, 2-4µm: Blue, 4-8µm: Yellow, 8-16µm: Red, 16-32µm: Green, 32-64µm: Cyan.

## 5    Summary and Conclusions

Dust aerosol deposition measurements by means of deposition and other passive samplers were

conducted at Izaña Global Atmospheric Watch observatory continuously from 14th of July to 24th of

August 2017. In addition, active aerosol collection was done with a free-wing impactor and an iso-axial

filter sampler. Additional information regarding the aerosol particle size distributions has been obtained

by an OPC Izaña. The single-particle data of size, flux and concentration of over 315,000 particles from

6 different samplers were obtained by applying a SEM-EDX technique. Different samplers are compared

based on size-resolved measurements, which makes our work unique when compared to previous works.

As known from previous studies, the total deposition flux was dominated by coarse particles (16-32 µm).

A high temporal variability is dust flux was observed on a daily basis.

The size resolved flux measurements of different passive samplers varied significantly between the

samplers under the same conditions. This is to be expected from the different sampler construction.

Applying suitable deposition velocity models, atmospheric concentrations can be calculated from



different sampler deposition fluxes, which are more in agreement. However, discrepancies beyond the
measurement uncertainty remain unexplained by the deposition models.
In particular when considering the size-resolved deposition velocities and flux ratios, great discrepancies
show up. While for an integrated bulk measurement or the $PM_{10}$ size range at least a qualitative agreement
can be reached, no model – analytical nor CFD – is able to explain the observed variations in deposition
flux between the samplers. Clearly, a better physical understanding is needed here.
The deposition velocity calculated from different models and flux measured from different samplers are
then used to calculate size resolved concentration. Nevertheless, the estimation of an appropriate
deposition velocity from different models is one of the main challenge of this work. The deposition
velocity model we applied in the calculation for concentration contains the gravitational and inertial
components of particle deposition.
We found also that the mass concentrations size distribution calculated from different passive samplers
have approximately the same values, which further confirms that the deposition velocity models selected
for this work are the appropriate ones to calculate mass concentration from mass flux. In this connection,
a comparison of friction velocity estimated from different approaches demonstrates that one approach
for some days is more pronounced than other measurement days, which could mean that the concentration
estimation from deposition flux might work better for a particular day with one approach than with the
other approach.
A very good agreement is found between the calculated concentration for samples from different passive
and active samplers and the concentration measured using OPC (Grimm) (this is particularly for particles
approximately in $PM_{10}$ size range). For particle sizes above $PM_{10}$, comparison of size distribution is made
to a novel FWI and comparison shows the results does not agree.
A deposition velocity results from different classical deposition models for different samplers are
compared to the deposition velocity calculated using a computational fluid dynamics simulations. The
comparison shows two methods do not agree. The deposition velocity calculated from computational
fluid dynamics looks more extreme in comparison to the one calculated from classical deposition models.
The correlation analysis between dust flux, dust concentrations and wind speed reveals that the change
in flux is mainly controlled by changes in concentration; variation of wind speed play a minor role for
wind speeds lower than 6 m/s. Situation might be different for higher wind speeds (e.g., Kandler et al.
2018). In connection to this, correlation analysis on number concentration calculated for samples from
different samplers yielded diverging results. It demonstrated that BSNE can be a good option for $PM_{10}$



measurement while MWAC as a horizontal flux sampler is not a suitable option for PM$_{10}$ measurement.
The analysis also showed that Flat plate and Sigma-2 geometries can also be a good option for measuring
PM$_{10}$ (Sigma-2 is better than Flat plate).
This data set provides the size-resolved information on deposition rate and concentration of mineral
aerosol particles which will help to assess special and temporal variability. A hypothesis of our study was
that the passive samplers could be capable of measuring size resolved particle concentration above the
PM$_{10}$ size range. However the results show that the samplers are not capable of producing measurements
consistent between the samplers or versus active collection techniques. Therefore, a recommendation
must be that if a certain sampler type is chosen for a study, it should not be modified or replaced by
another one for consistency of results. In a broader context, the results show nevertheless that passive
sampling techniques coupled with an automated single particle analysis provides insights into the
variation of size distribution, flux and concentration of atmospheric particles.














## 6    Acknowledgements


This project is funded by the Deutsche Forschungsgemeinschaft (DFG, German Research Foundation) –
264907654; 264912134; 416816480 (KA 2280). We would like to thank for the financial support by the
DFG in the framework of the Excellence Initiative, Darmstadt Graduate School of Excellence Energy
Science and Engineering (GSC 1070). We thank our colleagues Thomas Dirsch and Conrad
Ballschmiede. We are grateful to all staff members of Izaña Global Atmospheric Watch Observatory for
helping us in maintenance of the sampling equipment. We are especially indepted to Dr Roger Funk from
Leibniz-Centre for Agricultural Landscape Research, Institute of Soil Landscape Research for providing
us some of the passive samplers.

## 7    Author contribution

A. W. conducted the field measurements and the data evaluation. K. S. helped with the field
measurements, carried out the SEM analyses and did data processing. J. M. and B. E. executed the CFD
model setup and calculations. S. R. operated the OPC including the data processing and the
meteorological base measurements. K. K. designed the experiment, designed and prepared the sampling
equipment and did data processing and interpretation. All authors contributed to the data discussion and
manuscript preparation.

## 8    Data availability

The data sets used for this publication are available from the Pangaea repository free of charge (Waza et
al., doi: tbd.)










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
