# Peer review of "Field comparison of dry deposition samplers for collection of atmospheric"

_Atmospheric Measurement Techniques, 2019_

## Referee Comment (RC1) · Anonymous Referee #1 · 22 Jul 2019

This paper presents a comparison in between different four passive samplers that are commonly used for dust deposition measurements. In order to compare how these instruments perform, the authors have used a set of techniques (SEM, classic deposition velocity models, computational fluid dynamic simulations and other aerosol sampling instruments) that allow them to identify biases in these samplers. Although the used techniques are robust and the presented data is very relevant and will be very useful for the mineral dust community, I would only recommend this paper for publications after many major changes are addressed.

In spite of the good quality of the experimental work, at the present moment, the

manuscript seems to be in a very early stage of the publication process. There are many inconsistencies, typos, unexplained data and figures and it is extremely difficult to follow, particularly in the results and discussion section. In addition, some of the given conclusions are not really supported by the data, or they do in a vague way.

General comments.

One of the main problems of the manuscript is that the shown data is not well explained. It is not obvious for the reader understand how the data in each plot has been calculated. Sometimes, this information can be inferred from reading carefully the caption and the references to the figure in the text but this is not always the case and it makes it difficult to read the manuscript. See specific comments. In addition, the manuscript presents a large amount of data in different figures and tables but in some occasions, the discussion of these data is too short.

Lack of consistency. The magnitudes and concepts that appear through the text are mentioned in different ways, which makes the reading process very confusing. There are many other inconsistencies, such as the fact that some multi panels are not properly labelled using letters. See specific comments. In addition, there are many formatting issues. Some of them are pointed in the specific comments.

The geometry and computational fluid dynamics analysis of 3 passive samplers is given in the section 3. However, there are no references of the BSNE sampler in this section, while in the other sections, the four passive samplers have been mentioned. The geometry and computational fluid analysis of this fourth sampler should be included or at least justify its absence.

Many comparisons are presented all over the manuscript, but it seems that a significant fraction of the data hasn't been plotted and they appear instead in tables in the SI. I suggest to plot all the data that appears in tables in the SI. Some of the given conclusions regarding to the agreement or disagreement of data need to be revised. See specific comments.

In this manuscript many comparisons in between different instruments are presented. Were the sampling times of each instrument overlapping in all the cases? This remains unexplained, and it seems very unlikely in some occasions, instruments were ran with very different times (24h data compared with 1h data). See specific comments.

Regarding to the SEM analysis, were handling blanks taken during the campaign and then analysed under the SEM? In addition, did you test if the particles homogenously distributed over the sampling substrate? If not, this might significantly affect the measurements.

Specific comments.

Line 17. "This study focuses on the microphysical properties". This is too vague.

Line 19-32. This paragraph of the abstract looks more like a collection of statements that are made through the paper rather than a paper abstract.

Line 20. Acronyms in the abstract have not been defined before.

Line 26-28. Acronyms defined after they appear for the first time.

Line 97-98. What about the sampling time of the Flat plate sampler? Were the filters ran for one hour or 24 with the passive samplers?

Line 149. This section needs a bit more of detail.

Line 159. The acronym SMPS hasn't been defined. In addition, there are no other references to the SMPS in the main text. Was the data used for this work?

Line 164. How were the samples transported and stored?

Line 170. "Randomly selected areas". Were they randomly generated or were they selected manually by the user?

Line 194 and 222. Was the temperature dependence considered in the density and dynamic viscosity choice?

Line 258. I think this section needs to be better explained and describe why and how different models were applied to different samplers.

Line 423. Which was the fraction of mineral dust in the samples? Was it dominating all the sizes? Were the non mineral dust particles excluded from the calculations?

Line 430. In the mentioned tables, the size distribution for each collected sample is given in both mass flux and number flux. Why has it been described as "Minimum, Maximum and Median Mass Flux (mg/(m2d)) measured by..." in the captions of the table S1, S2, S3, S4, S5 and S6?

Line 431. Has all the data in this section been calculated with the SEM? If so indicate. It would be useful to also indicate it in the figure captions.

Line 435. In this section, the terms "deposition flux" and "mass flux" seem to be used to refer to the same magnitude. If this is the case, use only one notation, and mention alternative notations when the magnitude is introduced first.

Line 449. "we can clearly see that that there is high temporal variation in deposition flux between dust event days and non-dust event days". Fig. 9 doesn't clearly show this. There is a significant difference for the MWAC sampler, but for the other 3 instruments, the difference doesn't seem "high" for the first four bins (up to a factor 2-3?). This is difficult to see since there are not minor ticks in the y-axis. I suggest to add minor ticks and lines as for the x-axis, as well as softening the statement and explaining better the difference in between the deposition flux during a dust event and a non-dust event.

Line 450. "Generally, the temporal variation is much higher than difference between samplers". This statement seems a bit weak for the reasons mentioned previously (Line 449). In addition, all the data in the tables S1, S2, S3 and S4 hasn't been plotted so, it is difficult to see if these argument is valid for all the data. I think this should be improved by adding more graphs (maybe in the SI) or doing some systematic statistical analysis. Improve this

Line 451. As mentioned before, I suggest to add some y-axis minor ticks or plot it again in a way that allows the reader to understand the differences in the mass fluxes. This has been done for most of the other figures of the manuscript. Reducing the range y-axis range to 10-1 to 104 mg/(m2d) (there is no data at all in the 10-4 to 10-1 mg/(m2d) range) could help to better appreciate the differences between the different curves. Also, explain why some large size bins have been removed (is it due to a small number of particles in those bins?).

Line 456-485. In this section, the ratios in between magnitudes obtained with four different instruments have been compared. Why has the Sigma-2 instrument been used as the reference instrument? This section doesn't compare the other instruments within themselves at all. Why? I suggest to add some information about how the other instruments compare to each other or justify why this comparison has been omitted.

Line 473. Having a legend in order to identify the different days could help to under-stand or discuss why the ratios change that much from one day to each other. Why is the ratio in between the Flat plate and the Sigma-2 of the cyan blue day that low when compared with other days?

Line 474-485. It is very difficult to follow what has been plotted in Fig. 12. Is the blue data the mean ratio between each sampler and the Sigma-2 (same ratios as in the previous section but using number instead of mass)? Has the BSNE deposition velocity ratio modelled data been obtained with the Piskunov model as stated in line 259? The y-axis is labelled as deposition velocity ratio, however, ratios of dry deposition flux has been plotted as well. Are these ratios equivalent as one would expect from the equation 7? In general, I think that this figure and what has been plotted in it needs to be much better explained than it is now.

Line 475. "The deposition velocity ratio from models is often higher than the ratios derived from the mass and number". Is this something that happens in general and has been reported in other studies or does it only happen here? In the first case, add

some references.

Line 487. What has exactly been plotted in Fig. S2? It is not obvious from the description. Explain this properly.

Line 489-491. The anti-correlation reported by the authors in the number flux-wind speed data cannot be seen in Fig. S2. Remove this or justify based in some quantitative statistical analysis.

Line 496. Again, two different notations for the dust deposition flux have been used. How is this data related to Fig. S2?

Line 499. What has it been shown in the table 2? From the caption, the reader can understand than dust deposition flux (probably SEM measured) has been correlated to the external measurements of OPC particle number and wind speed. However, in the line 499 the authors suggest that the data in table 2 is a comparison in between the OPC measured concentration and the modelled concentration (using the models on the SEM flux data to obtain this concentrations?). This section is very confusing and unclear and it needs to be much better explained.

Line 503. Do you mean from the correlations in Table 2? If so indicate it. In the description of the table it says that the flux was correlated with OPC number concentration, but here the authors mention here PM10. Do you mean number concentration below 10 $\mu$m?

Line 513. How have you plotted the wind speed? Did you divide each day in 30 minute interval averages and then calculated the mean and standard deviation from this data (I guess 48 points per day)? Explain it in the figure caption.

Line 514. What are the blue boxes showing? Is it the 25 and 75 percentiles? Are the black vertical lines showing only one standard deviation?

Line 516. "Small particle". Is this a common notation in dust deposition studies to refer to the 1-10 $\mu$m size range?

Line 520-521. "The effect of wind speed on deposition velocity is negligible". Why?

Line 522. In the text, the apparent deposition velocity concept has been introduced as the ratio of the number flux to number concentration. I suggest to use deposition velocity in the y-axis label.

Line 528-529. "Mass concentrations calculated from different passive samplers agree generally well with respect to the statistical uncertainties". This agreement is not fully true for the July 28 and August 21 cases shown in Fig S1. Why?

Line 532. Isn't the mas flux example given here the same as in Fig. 9a but with a different y-axis scale? If so, choose another example. Another idea would be removing the whole section and discussing the consistency between samples in a previous section.

Line 532. Why does the max flux data measured by the MWAC differ so much from the others but when converting it to mass concentration it agrees with them? The deposition velocity has been calculated with the same model for the MWAC, BSNE and Flat plate.

Line 532. What is "impaction curve & Piskunov" in the legend? The concept of "impaction curve" hasn't been mentioned before.

Line 540. How were the number size distributions calculated from the flux measurements? This should be better explained here or in the caption.

Line 544. Why have these specific samples (and these specific instruments) were chosen as an example? I assume there are lots of potential comparisons (you sampled during many days with four different instruments). How do other samples taken in other days and/or with other instruments compare the the OPC measurements? It seems too arbitrary to show only 4 comparisons out of many and extract some generalist conclusions.

Line 545. This caption needs to be rewritten in a more clear way. Were the SEM obtained mass flux distribution converted into mass size distributions using the different

approaches and then transformed into number size distributions using a density value?.

Line 550. The Momentum flux approach data looks black not green.

Line 555. "the above figure (Figure 16)" should be referred as Figure 16 or Fig. 16.

Line 555-560. "also show the comparison of the mass concentration size distribution measurement". Fig. 16 doesn't show any mass size distribution. Please correct or explain this.

Line 563. In order to calculate the mass concentration measured by each sampler, don't you have to use the SEM obtained mass flux measurement and assume one of the mentioned models? You haven't mentioned yet a direct method to measure mass concentrations from the passive samplers.

Line 563. When were these samples collected? Why only 2 samples were shown?

Line 578. In the methods section, the authors indicate that the sampling time for the passive samplers was about 24 hours while for the FWI was only half an hour hour. Why have you plotted data that has been collected in such a different time interval?

Line 563. It is very difficult to see the y-axis scale. Could you add some minor ticks?

Line 569-571. What could be causing the disagreement at large sizes?

Line 578. Have you used a model to calculate the mass concentration from the mass flux measurements and then transformed this to number concentration?

Line 578. Why only BSNE measurements have been shown? Are they representative of the other passive samplers?

Line 578. In the methods section, the authors indicate that the sampling time for the passive samplers was about 24 hours while for the filter samples was only one hour. Why have you plotted data that has been collected in such a different time interval?

Line 581. Was the data in this section obtained following the same SEM approach as

for the flat plate sampler? This measurements need to be described more precisely.

Line 595. When were this samples taken?

Line 599. As mentioned before, why hasn't the BSNE included in this analysis? Explain.

Line 609. It is difficult to see agreement in between the Stokes model and the CFD for the MWAC sampler in Fig. 20.

Line 623. Why haven't the errors been propagated?

Line 626. What do the vertical clusters of data mean? Why are there so many measurements aligned? (Particularly in the d, e and f case).

Line 627. Has all the collected data been presented here?

Line 645-646. "atmospheric concentrations can be calculated from different sampler deposition fluxes, which are more in agreement". The statement about the increase in the agreement is a bit vague. In addition, it seems that only a subset of all the possible atmospheric concentration samples has been shown.

Line 648-649. "In particular when considering the size-resolved deposition velocities and flux ratios, great discrepancies show up". More detail in which deposition velocities and flux ratios is needed here.

Line 652-656. This paragraph describes again about the size-resolved concentration. Reduce it and merge it with the first paragraph that describes this (643-647).

Line 664-667. It seems that not all the data has been shown, therefore the reader cannot check this conclusion.

Technical corrections.

Line 205. "Ati".There is an "i" after the t in the denominator of the equation. The is missing a p if it is referring to particle density.

Line 207. Spaces must be included between number and unit (e.g. 2-4 $\mu$m).

Line 227. Does u-s means us? I suggest use the same notation.

Line 234. Units appear in the exponential notation in some occasions but in some others they don't. I suggest to use the exponential notation through the whole manuscript (m/s should be written as m s-1).

Line 236. "Wood1981". Wood 1981.

Line 351. Missing coma or full stop.

Line 514. Two notations have been used to describe the observatory. Through most of the text, "Izaña Global Atmospheric Watch observatory" has been used, but here, a different one has been used. You can mention both at the beginning and then use only one trough the text.

Line 581. "upward/downward-facing measurements"

Line 584. "Up-ward" and "Down-ward". Is this the right notation or is it upward and downward?

Line 603. "V-dp" was referred earlier in the paper as Vd. Use a consistent notation

Line 642. "variability of dust".

General: Missing a, b, c... labelling in the multi panels. Sub-indexes haven't been written in many figures (E.g. u-s instead of us).

---

## Referee Comment (RC2) · Mingjin Tang (Referee) · 30 Aug 2019

Dry deposition, as an important processes which determines lifetimes and impacts of aerosol particles, is not well understood, and one reason is that measurement of aerosol dry deposition is difficult and suffers from large uncertainties. Waza et al. collected mineral dust particles using several dry deposition samples and characterized these particles using SEM, and carried out CFD simulations to further understand the performance of these samplers. The topic is clearly very important, and the work has been carried out in a comprehensive manner. However, the manuscript is not well written, tedious and hard to follow. Overall I feel it is more like a chapter (or a few chapters) of a thesis. The manuscript should be restructured and edited before it is recommended for final publication. I give a few example below to show how I think the manuscript can be improved.

I feel there are too many figures used in the manuscript. For example, Fig. 4-7 present CFD results, and do they have to be included in the manuscript? Can some of them be moved into the supplement? There are only three points in Fig. 8, and so is it necessary to have this figure?

Line 457-460: Figure 13 is mentioned before Figure 10, and it took me a while to find Figure 13. In addition, can Fig. 13 be moved to supplement?

Line 553-554, line 651, line 641-642, and etc.: Quite frequently there are paragraphs which contain 1-2 short sentences. This makes the manuscript very fragmented and hard to follow.

Line 38: The paper by Jickells et al. (2005) in fact discusses the effect of mineral dust on oceanic biogeochemistry and thus should not be cited here. Instead, it should be cited in line 40-41. In addition, please consider citing new references instead of papers which were published >20 years ago.

Line 13: change "As result" to "As a result"; Line 42: change "is ranging" to "ranging"; Line 56: change "on addition" to "in addition"; line 84: change "station shields" to "station, shields". There are many grammatical errors and awkward sentences in the manuscript, and careful editing of the manuscript is needed.

---

## Author Comment (AC1) · 30 Sep 2019

Dear Referee #1, We thank you for the critical comments and suggestions to improve the manuscript (MS). We have considered the comments and modified the MS accordingly. Our detailed responses to the comments are given below. General comments: Referee's comment: One of the main problems of the manuscript is that the shown data is not well explained. It is not obvious for the reader understand how the data in each plot has been calculated. Sometimes, this information can be inferred from reading carefully the caption and the references to the figure in the text but this is not always the case and it makes it difficult to read the manuscript. See specific

comments. In addition, the manuscript presents a large amount of data in different figures and tables but in some occasions, the discussion of these data is too short. Authors' response: Major corrections in the revised MS are made. Moreover, answers given to specific comments can be seen in the specific comments section. Referee's comment: Lack of consistency. The magnitudes and concepts that appear through the text are mentioned in different ways, which makes the reading process very confusing. There are many other inconsistencies, such as the fact that some multi panels are not properly labelled using letters. See specific comments. In addition, there are many formatting issues. Some of them are pointed in the specific comments. Authors' response: Major corrections in the revised MS are made. Moreover, answers given to specific comments can be seen in the specific comments section. Referee's comment: The geometry and computational fluid dynamics analysis of 3 passive samplers is given in the section 3. However, there are no references of the BSNE sampler in this section, while in the other sections, the four passive samplers have been mentioned. The geometry and computational fluid analysis of this fourth sampler should be included or at least justify its absence. Authors' response: Because of resource limitation, we did the CFD analysis only for the three geometries (samplers) and therefore the BSNE was not included in the CFD analysis. Again, answer this specific referees' comment is given in the specific comment section. Referee's comment: Many comparisons are presented all over the manuscript, but it seems that a significant fraction of the data hasn't been plotted and they appear instead in tables in the SI. I suggest to plot all the data that appears in tables in the SI. Some of the given conclusions regarding to the agreement or disagreement of data need to be revised. See specific comments. Authors' response: We have now plotted for the whole campaign data showing comparison among samplers. Answers are given in the specific comment section. Referee's comment: In this manuscript many comparisons in between different instruments are presented. Were the sampling times of each instrument over lapping in all the cases? This remains unexplained, and it seems very unlikely in some occasions, instruments were ran with very different times (24h data compared with 1h data). See specific

comments. Authors' response: The reason why we set up different time interval is because one category of the sampler is operating passively while the other ones (the FWI and the Filter samplers) operate actively. The active ones have a much higher collection velocity. Therefore, we cannot set up the same time interval for both type of samplers, as this would result in either overloading of the active or underloading of one of the passive samplers. A further explanation is given under specific comments. However, based on the PM10 values recorded continuously, we have compared the active sampler interval with the passive samples one and found, that the average PM10 values of both intervals differ by 0.2 %. Therefore, we believe that the comparison is justified. Referee's comment: Regarding to the SEM analysis, were handling blanks taken during the campaign and then analysed under the SEM? In addition, did you test if the particles homogenously distributed over the sampling substrate? If not, this might significantly affect the measurements. Authors' response: Blank samples were analyzed. The contamination is small for the dust compounds (factor of 30-100 lower than the deposited particle numbers). A low density of pure iron particles is present, apparently already from the manufacturing process. These particles are identified by their chemical composition and removed from the dataset.

Specific comments: Referee's comment 1: Line 17. "This study focuses on the microphysical properties". This is too vague. Authors' response: A sentence is added in the revised MS to explain as clearly as possible. Referee's comment 2: Line 19-32. This paragraph of the abstract looks more like a collection of statements that are made through the paper rather than a paper abstract. Authors' response: Major corrections in the revised MS are made on abstract part. Referee's comment 3: Line 20. Acronyms in the abstract have not been defined before. Authors' response: Correction is made. Referee's comment 4: Line 26-28. Acronyms defined after they appear for the first time. Authors' response: Correction is made. Referee's comment 5: Line 97-98. What about the sampling time of the Flat plate sampler? Were the filters ran for one hour or 24 with the passive samplers? Authors' response: The sampling time for all passive sampler including the Flat plate sampler was set to be 24 hours. The filters ran for one

hour. Referee's comment 6: Line 149. This section needs a bit more of detail. Authors' response: More detail on the samples construction and principle is added. Referee's comment 7: Line 159. The acronym SMPS hasn't been defined. In addition, there are no other references to the SMPS in the main text. Was the data used for this work? Authors' response: We used data only from OPC. So, corrections in the revised MS are made. Referee's comment 8: Line 164. How were the samples transported and stored? Authors' response: All samples were stored in standard SEM storage boxes (Ted Pella Inc, Redding, CA, USA) in dry conditions at room temperature. Referee's comment 9: Line 170. "Randomly selected areas". Were they randomly generated or were they selected manually by the user? Authors' response: First, the user orients the microscope the circular deposition area and then the microscope selects smaller sub-areas randomly. Referee's comment 10: Line 194 and 222. Was the temperature dependence considered in the density and dynamic viscosity choice? Authors' response: We have used constant values for density and dynamic viscosity. It is already mentioned in the MS. Referee's comment 11: Line 258. I think this section needs to be better explained and describe why and how different models were applied to different samplers Authors' response: A more detailed explanation is added in section 2.9 (line 253-255 and line 258-259). Referee's comment 12: Line 423. Which was the fraction of mineral dust in the samples? Was it dominating all the sizes? Were the non-mineral dust particles excluded from the calculations? Authors' response: We found that the fraction of mineral dust in all samples were dominating in all size ranges (96 %) and therefore in calculation, we assumed the fraction of non-dust particles to be negligible. Referee's comment 13: Line 430. In the mentioned tables, the size distribution for each collected sample is given in both mass flux and number flux. Why has it been described as "Minimum, Maximum and Median Mass Flux (mg/(m2d)) measured by..." in the captions of the table S1, S2, S3, S4, S5 and S6? Authors' response: In the captions of the table S1, S2, S3, S4, S5 and S6, the unit '(mg/(m2d))' was used for mass flux (mass deposition rate) while the unit '1/(m2d)' was used for number flux (number deposition rate). Corrections are made in the revised electronic supplement.

Referee's comment 14: Line 431. Has all the data in this section been calculated with the SEM? If so indicate. It would be useful to also indicate it in the figure captions. Authors' response: All mass flux data in the section 4.1.1 is calculated with SEM. A sentence is added in the revised MS to make the information more clear. A sentence is added in the caption too. Referee's comment 15: Line 435. In this section, the terms "deposition flux" and "mass flux" seem to be used to refer to the same magnitude. If this is the case, use only one notation, and mention alternative notations when the magnitude is introduced first. Authors' response: Changed to mass deposition rate in the revised MS. Referee's comment 16: Line449. "we can clearly see that that there is high temporal variation in deposition flux between dust event days and non-dust event days". Fig. 9 doesn't clearly show this. There is a significant difference for the MWAC sampler, but for the other 3 instruments, the difference doesn't seem "high" for the first four bins (up to a factor 2-3?). This is difficult to see since there are not minor ticks in the y-axis. I suggest to add minor ticks and lines as for the x-axis, as well as softening the statement and explaining better the difference in between the deposition flux during a dust event and a non-dust event. Authors' response: The plot is modified in the revised MS. And the statement is modified according to the referee's comment. Referee's comment 17: Line450. Line 450. "Generally, the temporal variation is much higher than difference between samplers". This statement seems a bit weak for the reasons mentioned previously (Line 449). In addition, all the data in the tables S1, S2, S3 and S4 hasn't been plotted so, it is difficult to see if these argument is valid for all the data. I think this should be improved by adding more graphs (maybe in the SI) or doing some systematic statistical analysis. Improve this. Authors' response: a box plot showing temporal variation of size distribution is added to the revised Manuscript. Referee's comment 18: Line 451. As mentioned before, I suggest to add some y-axis minor ticks or plot it again in a way that allows the reader to understand the differences in the mass fluxes. This has been done for most of the other figures of the manuscript. Reducing the range y-axis range to 10-1 to 104 mg/(m2d) (there is no data at all in the 10-4 to 10-1 mg/(m2d) range) could help to better appre-

ciate the differences between the different curves. Also, explain why some large size bins have been removed (is it due to a small number of particles in those bins?). Authors' response: The graph is corrected in revised MS. Regarding the last data point, there was not data actively removed. Although particles across all size ranges (up to approximately 100 $\mu$m) can be deposited on the passive samplers, in our analysis we did generally not find particles larger than 64 $\mu$m diameter. When the last data point is missing from the plots, no particle between 32 and 64 $\mu$m was detected. Referee's comment 19: Line 456-485. In this section, the ratios in between magnitudes obtained with four different instruments have been compared. Why has the Sigma-2 instrument been used as the reference instrument? This section doesn't compare the other instruments within themselves at all. Why? I suggest to add some information about how the other instruments compare to each other or justify why this comparison has been omitted. Authors' response: The sampler Sigma-2 has been widely used for deposition sampling and therefore, in this work it is used as reference. A comparison showing other samplers as reference is added (see electronic supplement). Referee's comment 20: Line 473. Having a legend in order to identify the different days could help to understand or discuss why the ratios change that much from one day to each other. Why is the ratio in between the Flat plate and the Sigma-2 of the cyan blue day that low when compared with other days? Authors' response: A legend is already added to different measurement days to identify the different days and can be seen in the revised MS. The flux ratio of Flat plate to Sigma-2 of the cyan blue day (July 29, 2017) is low when compared with other days. The low value of the deposition rate observed with flat plate for this particular day cannot be explained by other observations, so it has to be considered as an artifact. Therefore, we show the data, but we do not take it into account for further discussion. Referee's comment 21: Line 474-485. It is very difiňĄcult to follow what has been plotted in Fig. 12. Is the blue data the mean ratio between each sampler and the Sigma-2 (same ratios as in the previous section but using number instead of mass)? Has the BSNE deposition velocity ratio modelled data been obtained with the Piskunov model as stated in line 259? They-axis is labelled as deposition velocity ratio, however, ratios of dry deposition flux has been plotted as well. Are these ratios equivalent as one would expect from the equation 7? In general, I think that this figure and what has been plotted in it needs to be much better explained than it is now. Authors' response: Generally, Fig. 12 shows a comparison of velocity ratios of sampler A to Sampler B obtained from flux measurement to the velocity ratio obtained from different classical deposition velocity models. The ratio of flux measured by one sampler to flux measured by another sampler is equal to the velocity ratio of the two sampler. In the Fig. 12., the blue data shows the velocity ratio obtained from flux measurement while the red one shows the velocity ratios obtained from different deposition velocity models. A table showing different deposition velocity models used for different samplers is added in revised MS in section 2.9.3 (see Table 1). The flux ratios plotted in the Fig. 10 and 11 is meant to show the relative collection efficiencies of different sampler with respect to reference sampler (Sigma-2). The paragraph has been rewritten to clarify the type of display. Referee's comment 22: Line 475. "The deposition velocity ratio from models is often higher than the ratios derived from the mass and number". Is this something that happens in general and has been reported in other studies or does it only happen here? In the first case, add some references. Authors' response: It is true that the deposition velocity ratio from models is higher than the ratios derived from the mass and number flux. We are not aware of any other studies that have been done on the subject. However, as this is only a relative display, there cannot be any 'truth' (most accurate sampler) derived. The higher ration can mean an underestimation of the Sima-2 deposition velocity, or an overestimation of the others. This is stated now in the manuscript. Reviewer comment: Line 487. What has exactly been plotted in Fig. S2? It is not obvious from the description. Explain this properly. The main purpose of section 4.1.2 to investigate the driving force of atmospheric deposition rate. As already stated in section 4.1.2, Figure S 2 (now Figure S 10 in the revised electronic supplement) displays the correlation between deposition number fluxes (measured by flat plate sampler) and atmospheric number concentration by the OPC. An extended caption has been added to the figure. Referee's comment 23: Line 489-491. The anticorrelation reported by the authors in the number flux-wind speed data cannot be seen in Fig. S2. Remove this or justify based in some quantitative statistical analysis. Authors' response: It is correct, there is not significant correlation for wind speed in Fig. S2 (S10). Justification based on quantitative statistical analysis is added in the revised MS to make clear. In addition to the Fig. S2 (now Figure S 10 in the revised electronic supplement), a quantitative statistical analysis was already shown by Table 2 (now table 3). Referee's comment 24: Line 496. Again, two different notations for the dust deposition flux have been used. How is this data related to Fig. S2? Authors' response: Correction was made on notations for dust deposition flux. Generally, Fig. S2 and table 2 (now Table 3) shows the dependence of small particle dust deposition flux on atmospheric PM10 concentration and wind speed. While Fig S2 (now Figure S 10 in the revised electronic supplement) shows the correlation between flux, dust concentration, and wind speed for samples measured by flat plate sampler, table 2 (now Table 3 in the revised MS) (Line 496) shows the same relation using quantitative statistical analysis for all samplers (Flat plate, MWAC, BSNE, Sigma-2). Referee's comment 25: Line 499. What has it been shown in the table 2? From the caption, the reader can understand than dust deposition flux (probably SEM measured) has been correlated to the external measurements of OPC particle number and wind speed. However, in the line 499 the authors suggest that the data in table 2 is a comparison in between the OPC measured concentration and the modelled concentration (using the models on the SEM flux data to obtain this concentrations?). This section is very confusing and unclear and it needs to be much better explained. Authors' response: Line 499-500 in section 4.1.2 should not have referred to Table 2 (now table 3 in the revised MS). The authors replaced this table with Table S7. Table S7 shows a quantitative statistical analysis for correlation between the OPC measured concentration and the modelled concentration (using deposition velocity models on the SEM flux). In addition, the paragraph has been rewritten to clarify the approach. Referee's comment 26: Line503. Do you mean from the correlations inTable2? If so indicate it. In the description of the table it says that the flux was correlated with OPC number concentration, but here

the authors mention here PM10. Do you mean number concentration below 10 $\mu$m? Authors' response: In line 503, the correlations refer to Table 2 (now table 3 in the revised MS). The table shows the correlation between flux OPC number concentration in PM10 size range. A description is corrected on the table 2 (now table 3 in the revised MS). And also the size range is explicitly stated now at the beginning of section 4.1.2. Referee's comment 27: Line 513. How have you plotted the wind speed? Did you divide each day in 30-minute interval averages and then calculated the mean and standard deviation from this data (I guess 48 points per day)? Explain it in the figure caption. Authors' response: A 30-min averaged wind speed data was obtained by dividing each day data in 30-minute interval averages and then the mean and standard deviation was calculated from this data. An explanation was added into the figure caption in the revised MS. Referee's comment 28: Line 514. What are the blue boxes showing? Is it the 25 and 75 percentiles? Are the black vertical lines showing only one standard deviation? Authors' response: Yes. On each blue box, the central mark is the median, the edges of the box are the 25th and 75th percentiles. The black vertical lines show one standard deviation. An explanation was added into the figure caption in the revised MS. Referee's comment 29: Line 516. "Small particle". Is this a common notation in dust deposition studies to refer to the 1-10 $\mu$m size range? Authors' response: "Small particle" notation was used to refer to PM10 size range. Accordingly, the title of section 4.1.2.1 is changed to "Size-resolved apparent deposition velocity in the PM10 size range" in the revised MS. Referee's comment 30: Line 520-521. "The effect of wind speed on deposition velocity is negligible". Why? Authors' response: As already indicated by Table 2, there is not significant correlation between the wind speed and the observed deposition rate. While this could be still a second order effect of an anticorrelation between atmospheric concentration and wind speed, Fig. 14 shows clearly, that there is not wind speed effect for the smaller particles. While this is in contradiction to the models, one has to keep in mind that the (a) the observed wind speeds are comparatively low here, and (b) the considered size range is not the most affected. An effect of the wind speed might be much stronger at higher wind speed

and for larger particles. An according statement is added to the manuscript. Referee's comment 31: Line 522. In the text, the apparent deposition velocity concept has been introduced as the ratio of the number flux to number concentration. I suggest to use deposition velocity in the y-axis label. Authors' response: the y-axis label is changed to deposition velocity in the revised MS. Referee's comment 32: Line 528-529. "Mass concentrations calculated from different passive samplers agree generally well with respect to the statistical uncertainties". This agreement is not fully true for the July 28 and August 21 cases shown in Fig S1. Why? Authors' response: The agreement generally holds true with the respect to the mean value of the campaign. And yes, it is correct that agreement might not be true in single cases. Referee's comment 33: Line 532. Isn't the mas flux example given here the same as in Fig. 9a but with a different y-axis scale? If so, choose another example. Another idea would be removing the whole section and discussing the consistency between samples in a previous section. Authors' response: The authors are aware of the case that mas flux examples given in section 4.1.3.1 (Fig. 15) and the one in section 4.1.1 (Fig. 9a) are the same, but they do have different message. The message of the Fig. 9a is to show the mass flux measured during dust event day differs from the one measured during non-dust event days (Fig. 9b). The other message of Fig 9a is to show the variation in mass flux measured by different passive samplers (for the same measurement day). The purpose of Fig. 15 is to show the consistence in concentration obtained from flux measurement for different samplers and to show that different deposition velocity models selected for the samplers are generally suitable, despite the deviations in single cases. In addition, more one more day is added to Fig. 15 in the revised MS. Referee's comment 34: Line 532. Why does the max flux data measured by the MWAC differ so much from the others but when converting it to mass concentration it agrees with them? The deposition velocity has been calculated with the same model for the MWAC, BSNE and Flat plate. Authors' response: This seems to be a misunderstanding. MWAC is calculated with the different velocity model (shown in Table 2 now). Therefore, the model the observed differences in deposition rate to a similar range concentration comparatively

well. Deposition velocity used for different samplers is explicitly indicated in section 2.9.3 (see table 1 in the revised MS). Referee's comment 35: Line 532. What is "impaction curve & Piskunov" in the legend? The concept of "impaction curve" hasn't been mentioned before. Authors' response: The impaction curve was briefly introduced in the method section 2.9.3. To clarify, section 2.9.3 has been reworked and Table 1 added. Referee's comment 36: Line 540. How were the number size distributions calculated from the flux measurements? This should be better explained here or in the caption. Authors' response: To get the number concentration size distributions, first the number flux (#/(m2day)) measured by different samplers is obtained from SEM. Then, the SEM number flux is converted in to number concentration by using different deposition velocity models. An explanation on how number concentration size distribution is calculated is added in the caption in the revised MS. Referee's comment 37: Line 544. Why have these specific samples (and these specific instruments) were chosen as an example I assume there are lots of potential comparisons (you sampled during many days with four different instruments). How do other samples taken in other days and/or with other instruments compare the OPC measurements? It seems too arbitrary to show only 4 comparisons out of many and extract some generalist conclusions. Authors' response: More samples (representing dust event days and non-dust days) are added (see revised electronic supplement). These specific samples shown in the figure (in the MS) are exemplary and they represent a particular dust event day. However, more comparison involving this section can be obtained in the electronic supplement (randomly selected from dust event day and non-dust event day from all samplers. Referee's comment 38: Line 545. This caption needs to be rewritten in a more clear way. Were the SEM obtained mass flux distribution converted into mass size distributions using the different approaches and then transformed into number size distributions using a density value? Authors' response: An explanation was given in the previous referee's comment (Referee's comment 36: Line 544). Caption was changed accordingly. Referee's comment 39: Line 550. The Momentum flux approach data looks black not green Authors' response: 'The Momentum flux approach' data is

changed from green to black in the revised MS. Referee's comment 40: Line 555. "the above figure (Figure 16)" should be referred as Figure 16 or Fig. 16. Authors' response: "the above figure (Figure 16)" is changed to Figure 16 in the revised MS. Referee's comment 41: Line 555-560. "also show the comparison of the mass concentration size distribution measurement". Fig. 16 doesn't show any mass size distribution. Please correct or explain this. Authors' response: In "also show the comparison of the mass concentration size distribution measurement" sentence, 'mass concentration' is replaced by, 'number concentration' in the revised MS. Referee's comment 42: Line 563. In order to calculate the mass concentration measured by each sampler, don't you have to use the SEM obtained mass flux measurement and assume one of the mentioned models? You haven't mentioned yet a direct method to measure mass concentrations from the passive samplers. Authors' response: This is correct; we have added a clarification to the caption in the revised MS. Referee's comment 43: Line 563. When were these samples collected? Why only 2 samples were shown? Authors' response: The purpose of the figure is to show the comparison of concentration measured by different passive samples with that concentration measured by active sampler (FWI) and OPC. Concentration measured by passive samplers through the campaign (see the electronic supplement). We could do an ESEM analysis only for four days' samples from FWI (which is a total of 12 samples) (from July 26, 2017 to July 29, 2017; each day three measurements) due to limited resources. So we compared the available measurements from passive samples with that of FWI for only of those four days. The two-day measurements (samples) shown in the figure are arbitrary taken examples and are daily average measurements. They were collected on 26th of July and 27th of July. The information was added in the caption. The authors have analyzed a total of 6 samples from FWI on 26th of July and 27th of July. A clarification is added in the caption in the revised MS. The comparison for all 4 days for FWI yields the same behavior. In addition, in the revised electronic supplement, comparison with remaining two analyzed days of FWI samples are shown. Referee's comment 44: Line 578. In the methods section, the authors indicate that the sampling time for the passive samplers

was about 24 hours while for the FWI was only half an hour. Why have you plotted data that has been collected in such a different time interval? Authors' response: The reason why we set up different time interval is because one category of the sampler is operating passively while the other one, which is, a FWI operates actively. Therefor we cannot set up the same time interval for both types of samplers. FWI as an active sampler needs less time than the passive ones. However, we calculated from OPC the average PM10 for the hours of the FWI samplings and compare it with the PM10 of the respective deposition samplings from OPC and we found that the average PM10 values of both intervals differ by 2 %. Therefore, we think it is justified to compare samples from FWI and other passive samplers collected with different time interval. Referee's comment 45: Line 563. It is very difficult to see the y-axis scale. Could you add some minor ticks? Authors' response: The minor ticks are added in the y-axis scale (see the revised MS). Referee's comment 46: Line 569-571. What could be causing the disagreement at large sizes? Authors' response: This is more of a speculation, but the FWIs inherently don't have an inlet at all, whereas all of the passive samplers have an inlet like structure, so the large particles might not be able to enter the inlet, when due to the atmospheric wind direction fluctuations the wind vector is not in parallel with the inlet axis. Also other types of inlet losses in the growing boundary layer might occur, which are not regarded by the models. We have added a cautious sentence on that. Referee's comment 47: Line 578. Have you used a model to calculate the mass concentration from the mass flux measurements and then transformed this to number concentration? Authors' response: From SEM measurements, both the number and mass deposition rate are obtained for each single particle. So the same size-resolved model can be applied to convert the deposition rates into number size concentrations. We have added an explicit statement to the method section 2.9. Referee's comment 48: Line 578. Why only BSNE measurements have been shown? Are they representative of the other passive samplers? Authors' response: In section 4.1.2 (table 2 (now table 3 in the revised MS)), the authors showed that BSNE is actually a suitable instrument for a PM10 estimation. In this connection, the authors showed

the comparison of number concentration measured with Filter-sampler method, BSNE and OPC. Measurements by other samplers are shown in the electronic supplement. Referee's comment 49: Line 578. In the methods section, the authors indicate that the sampling time for the passive samplers was about 24 hours while for the filter samples was only one hour. Why have you plotted data that has been collected in such a different time interval? Authors' response: Please refer to the answer to comment 44. Referee's comment 50: Line 581. Was the data in this section obtained following the same SEM approach as for the flat plate sampler? These measurements need to be described more precisely. Authors' response: The same SEM approach has been used in this section also. The only difference is that the flat plate geometry with 25mm-stub used here to collect particles where as in the flat plate described in the previous section, the stub was 12mm size was used. Precise description for the upward and downward flux measurement has been indicated in the section 2.4 in the revised MS. Referee's comment 51: Line 595. When were this samples taken? Authors' response: A legend is added to show different sampling dates (see the revised MS). Referee's comment 52: Line 599. As mentioned before, why hasn't the BSNE included in this analysis? Explain. Authors' response: Due to resource limitations, please refer to the comment above. Referee's comment 53: Line 609. It is difficult to see agreement in between the Stokes model and the CFD for the MWAC sampler in Fig. 20. Authors' response: Indeed, the agreement is poor in general. Regarding the mentioned models and sampler, this appears to be a misunderstanding regarding the 'general agreement'. it has been rephrased. Referee's comment 54: Line 623. Why haven't the errors been propagated? Authors' response: Refer to the revised MS for explanation. Referee's comment 55: Line 626. What do the vertical clusters of data mean? Why are there so many measurements aligned? (Particularly in the d, e and f case). Authors' response: The vertical clusters of data mean that for different wind speed situations, similar ratios are measured, where the models would predict different ratios. This was already seen above, where in contrast to the model prediction, no wind speed dependence was observed. Referee's comment 56: Line 627. Has

all the collected data been presented here Authors' response: All collected data (i.e. simultaneously analyzed samples from different samplers) has been analyzed and is show here. Referee's comment 57: Line 645-646. "atmospheric concentrations can be calculated from different sampler deposition fluxes, which are more in agreement". The statement about the increase in the agreement is a bit vague. In addition, it seems that only a subset of all the possible atmospheric concentration samples has been shown Authors' response: More data (campaign average) is added and the plot can be seen in the revised electronic supplement. The samplers are better in agreement with respect to the average, when the models are employed to calculate the concentration, but temporal variation correlation does not get better. Referee's comment 58: Line 648-649. "In particular when considering the size-resolved deposition velocities and flux ratios, great discrepancies show up". More detail in which deposition velocities and flux ratios is needed here. Authors' response: See the revised MS for explanation. Referee's comment 59: Line 652-656. This paragraph describes again about the size-resolved concentration. Reduce it and merge it with the first paragraph that describes this (643-647). Authors' response: The paragraph is reduced and merged into line 643-647 (see the revised MS). Referee's comment 60: Line 664-667. It seems that not all the data has been shown, therefore the reader cannot check this conclusion Authors' response: More data is shown now in the revised MS. Technical corrections Referee's comment 61: Line 205. "Ati".There is an "i" after the t in the denominator of the equation. The is missing a p if it is referring to particle density. Authors' response: Correction is made. Referee's comment 62: Line 207. Spaces must be included between number and unit (e.g. 2-4 $\mu$m). Authors' response: Correction is made in the revised MS. Referee's comment 63: Line 227. Does u-s mean us? I suggest use the same notation. Authors' response: Correction is made in the revised MS. Referee's comment 64: Line 234. Units appear in the exponential notation in some occasions but in some others they don't. I suggest to use the exponential notation through the whole manuscript (m/s should be written as m s-1) Authors' response: Majority of the exponential notation Units through the whole manuscript is written in

the form of 'a/b' and therefore we changed the units from exponential notation to the 'a/b' form though out the manuscript in the revised MS. Referee's comment 65: Line 236. "Wood1981". Wood 1981. Authors' response: Correction is made in the revised MS. Referee's comment 66: Line 351. Missing coma or full stop Authors' response: Correction is made in the revised MS. Referee's comment 67: Line 514. Two notations have been used to describe the observatory. Through most of the text, "Izaña Global Atmospheric Watch observatory" has been used, but here, a different one has been used. You can mention both at the beginning and then use only one trough the text. Authors' response: Correction is made in the revised MS. Referee's comment 68: Line 581. "upward/downward-facing measurements" Authors' response: We do not understand this comment. The collection surface in this measurement is facing to each other in upward-downward direction and thus the name "upward/downward-facing measurements" is used. Referee's comment 69: Line 584. "Up-ward" and "Down-ward". Is this the right notation or is it upward and downward? Authors' response: "Up-ward" and "Down-ward" is replaced by "upward and downward" in the revised MS. Referee's comment 70: Line 603. "V-dp" was referred earlier in the paper as Vd. Use a consistent notation Authors' response: "V-dp" is replaced by "Vd" in the revised MS. Referee's comment 71: Line 642. "variability of dust". Authors' response: We do not understand this comment. General Referee's comment 72: Missing a, b, c... labelling in the multi panels. Sub-indexes haven't been written in many iňĄgures (E.g. u-s instead of us). Authors' response: multi panels has been labelled with a, b, c. . . and indicated in the caption. Sub-indexes have been corrected now (see the revised MS).

The answers are all in the supplement!

Please also note the supplement to this comment:
https://www.atmos-meas-tech-discuss.net/amt-2019-187/amt-2019-187-AC1-supplement.zip

---

## Author Comment (AC2) · 30 Sep 2019

Dear Mingjin, We thank you for the critical comments and suggestions to improve the manuscript (MS). We have considered the comments and modified the MS accordingly. Our detailed responses to the comments are given below. General comments: For your general comments, major corrections in the revised MS are made. Moreover, answers given to specific comments can be seen in the specific comments section. Specific comments: Comment: I feel there are too many figures used in the manuscript. For example, Fig. 4-7 present CFD results, and do they have to be included in the manuscript? Can some of them be moved into the supplement? There

are only three points in Fig. 8, and so is it necessary to have this figure? Authors'
response: Some of the figures present in the CFD result (Fig. 4-7) are now moved to
electronic supplement (see the revised electronic supplement). In Fig. 8, we wanted to
show how the mean flow velocity in the MWAC tube varies as a function of the outside
velocity for the three cases. We have also moved Fig. 8 to the electronic supplement.
Comment: Line 457-460. Figure 13 is mentioned before Figure 10, and it took me a
while to find Figure 13. In addition, can Fig. 13 be moved to supplement? Authors'
response: The figure is now moved to the supplement. Comment: Line 553-554, line
651, line 641-642, and etc.: Quite frequently there are paragraphs which contain 1-
2 short sentences. This makes the manuscript very fragmented and hard to follow.
Authors' response: Correction is made in the revised MS. Comment: Line 38: The
paper by Jickells et al. (2005) in fact discusses the effect of mineral dust on oceanic
biogeochemistry and thus should not be cited here. Instead, it should be cited in line
40-41. In addition, please consider citing new references instead of papers which were
published >20 years ago. Authors' response: Correction is made in the revised MS.
Quite a few references we used are old indeed; but we cannot leave them as of our
formulations are based on them. Referee's comment: Line 13: change "As result" to
"As a result"; Line 42: change "is ranging" to "ranging"; Line 56: change "on addition"
to "in addition"; line 84: change "station shields" to "station, shields". There are many
grammatical errors and awkward sentences in the manuscript, and careful editing of
the manuscript is needed. Authors' response: Correction is made in the revised MS.

The answers are all in the supplement!

Please also note the supplement to this comment:
https://www.atmos-meas-tech-discuss.net/amt-2019-187/amt-2019-187-AC2-
supplement.zip

---

## Referee Report (RR1)

I would like to thank the authors for their detailed response to the referee's comments. The manuscript looks much better now, it has gained a lot of coherence, it is easier to read, more compact and with a better formatting. However, there are still some points to address before it is ready for publication.

Most of the comments of the previous review have been responded correctly and therefore accepted. However, there are some for which I still have some comments. In addition, I have some other comments and suggestions on other parts of the paper to address before it is ready for publication. See below:

- Referee's previous comment 37 and 60: In the previous version of the manuscript, only comparisons from a few days had been shown. Now, extra comparisons have been added to both the MS and the electronic supplement, which is good. Have all the possible comparisons been added? If not, I still understand that showing all the possible comparisons taken during the whole campaign could be excessive. However, I recommend to the authors that at least take a look on all the possible comparisons of the campaign and explain of the conclusions extracted from the shown samples are still valid. This is valid, in general whenever some comparisons are omitted.

- Referee's previous comment 48: I would add to the MS the reason why BSNE has been chosen.

- General comment on the samplers, FWI and filter comparisons: (line 580-610, related to referee's previous comment 44): The main caveat of the comparisons between the FWI vs deposition samplers and filter sampler vs deposition samplers is the different sampling time. I understand the need to use different sampling times for different instruments but the comparison should be justified, since in some cases you are comparing a 1 hour sample with a 24 hour one. It seems very surprising that PM10 values changed only up to 2 percent over a 24 hour period since there is a big variability (more than one order of magnitude) over the whole campaign for values such as the deposition rate or number concentration. As an example, in Fig. 13, the authors have shown the size distribution calculated from the filters on the 26th and 27th of July (I understand that they are two 1 hour samples separated 24 from each other) and the difference seems about a factor 2, which is not small. I strongly recommend to extend the analysis (maybe not in the main part of the paper but in the electronic supplement) on how the number concentration of different size bins change over the 24 hour averaging periods in order to justify that 1 hour samples can be compared with 24 hour samples.

- Sect 2.3: this is not very crucial but as a suggestion, I would organise it as follows:

2.3 Particle sampling

      2.3.1 Dust deposition samplers

            2.3.1.1 Flat plate sampler

            2.3.1.2 Sigma-2 sampler

            2.3.1.3 The Modified Wilson and Cooke (MWAC) sampler

            2.3.1.4 The Big Spring Number Eight (BSNE) sampler

>   2.3.2 Free-wing impactor (FWI)

>   2.3.3 Filter sampler

And I would add the Upward-downward sampler to the Sect 2.3 or 2.3.1 since they are essentially dust samplers that are analysed in the same approach.

- Sect 2.11. Could you add a short description of the inlet biases? From figure 11 it seems that they are negligible? If the differences are so small, is it worth it to show this correction?

- Table S1: I don't understand when it says "Minimum, maximum and median daily basis mass deposition rate". From what I see in the data, it seems that you have reported the mass deposition rate size distribution (average I guess) for each day, but I don't understand where the maximum and minimum are. This happens for other tables in the SI.

- Fig. 5 and Fig 6. Inconsistent notation in the axis labels

- Fig. 6. Maybe mention in the caption that the median, percentiles and standard deviations shown there correspond to the variability of the whole campaign for each instrument and bin.

- Line 464-466 "The deposition rate ratios obtained from the measurements are identical to the deposition velocity ratios, when the sampling time and concentration are the same". What does this mean? Which deposition velocity ratios do you mean in the second case?

- Lines 498-492. Table 3 shows that there is a positive dependency in between the concentrations and number deposition rates (r coefficients are positive, so if you increase one variable, the other also does) but it doesn't show a linear correlation since the r2 values are not close to 1 for the first two samplers (particularly the MWAC). However there is a good correlation for the last two samplers. Please clarify this in the text. Also, when it is written "see Figure S 16" it should be "see Figure S 16a".

- As a suggestion, you probably don't need to mention "in the electronic supplement" (e.g: "see Figure S X in the electronic supplement") every time since the label Figure S X already means electronic supplement.

- Line 524. "Negligible". Wouldn't it better to use another concept to describe the relation in between the wind speed and the deposition velocities? The analysis shown shows that the wind speed doesn't correlate with the deposition velocity and rate at all but it doesn't necessarily show the effect is negligible.

- Figure S 9 (and also previous referee comment 32). How can the errors of the campaign average be so high? For the MWAC, as an example, you are basically reporting a range of about 4.5 orders of magnitude for each bin. I suggest checking how errors have been calculated here and reconsidering if it is worth it to call that agreement considering the massive uncertainties.

- Line 555. "Overall, the number concentrations obtained from OPC measurements are slightly higher than the ones". This is not that clear, particularly for the Sigma-2. Please it discuss more.

- Line 560. Isn't his statement also supported by the Figure S11? If so indicate.

- Line 564. Legend of the panels a and c and d should be reordered: OPC, Momentum flux, Momentum flux PM10, Wood 1981 and Wood 1981 PM10. This would be make easier to realise that the difference in between applying the correction and not applying it is so small. In addition, I think the axis limits should be chosen differently since more than half of the graph is empty. (5 orders of magnitude are shown while the data only scatters over 2 orders of magnitude in the y axis).

- Line 569. "Deosition" should be deposition.

- Fig S11 and S12. Why not to split each of this figures into part a and part b, and having the multi panel with all the information for each sampler under the same title? In the caption of the Fig S11, yellow and green are mentioned while data has been plotted in red and green.

- Line 586. "The blue curve shows the concentration measurements by the OPC". This is not necessary since it appears in the legend.

- Figure S 10. Why is there such a massive discrepancy between the Sigma-2 and the rest of the instruments on the 29th of July? Also, the colour choice seems a bit arbitrary, why does cyan appears twice?

- Line 600. I would mention clearly somewhere that graphs like the Figure 13 but for the other samplers have been presented in the SI, as well as the reason why the BSNE has been shown primarily (see response to referee's pervious comment 48). Also, this section in general needs a bit more of discussion on the agreement or disagreement of the results. For example, the disagreement for the flat plate (I the supplement) is not as significant as the others.

- Line 601-602 "The reasons for this weak correlation – in particular in comparison to the ones 601 from Sigma-2 and BSNE – remain unexplained by now". Does this mean the reasons for the disagreement between the different samplers and the OPC/filters? If so explain better.

- Line 638-641. I would add some lines summarising the comparisons shown in the electronic supplement.

Other general comments:

- In some occasions, a few examples of a comparison are shown in the MS, while the bulk of the data is presented in the electronic supplement. (e.g: Fig 11, S11 and S12). I would state it more clearly in the main text for the sake of simplicity.

- Many figures, particularly in the electronic supplement, don't have a legend (and it is explained instead in the caption). In general it is better to have legends than explanations in the caption if possible.

---

## Author Response (AR2)

Dear Referee #1,

We thank you for the critical comments and suggestions to improve the manuscript (MS). We have considered the comments and modified the MS accordingly. Our detailed responses to the comments are given below.

Referee's comment- Referee's previous comment 37 and 60: In the previous version of the manuscript, only comparisons from a few days had been shown. Now, extra comparisons have been added to both the MS and the electronic supplement, which is good. Have all the possible comparisons been added? If not, I still understand that showing all the possible comparisons taken during the whole campaign could be excessive. However, I recommend to the authors that at least take a look on all the possible comparisons of the campaign and explain of the conclusions extracted from the shown samples are still valid. This is valid, in general whenever some comparisons are omitted.

Regarding Fig. 13 and Figs. S11-S16 (original Fig. 16), all possible comparisons are shown. The number of comparisons is lower as the total number of samples for each type, because of each sampler, some samples were not analyzable (contamination, sample loss). In particular, with respect to Sigma-2 and BSNE (Fig. S14 and S16) we still think that the comments on general good agreement are justified.

Referee's previous comment 48: I would add to the MS the reason why BSNE has been chosen.

Authors' response: The explanation is added.

Referee's comment- General comment on the samplers, FWI and filter comparisons: (line 580-610, related to referee's previous comment 44): The main caveat of the comparisons between the

FWI vs deposition samplers and filter sampler vs deposition samplers is the different sampling time. I understand the need to use different sampling times for different instruments but the comparison should be justified, since in some cases you are comparing a 1 hour sample with a 24 hour one. It seems very surprising that PM10 values changed only up to 2 percent over a 24 hour period since there is a big variability (more than one order of magnitude) over the whole campaign for values such as the deposition rate or number concentration. As an example, in Fig. 13, the authors have shown the size distribution calculated from the filters on the 26th and 27th of July (I understand that they are two 1 hour samples separated 24 from each other) and the difference seems about a factor 2, which is not small. I strongly recommend to extend the analysis (maybe not in the main part of the paper but in the electronic supplement) on how the number concentration of different size bins change over the 24 hour averaging periods in order to justify that 1 hour samples can be compared with 24 hour samples.

Authors' response: Analysis on how the number concentration of different size bins change over the 24 hour averaging periods is now included in the revised Supplement.

Referee's comment on- Sect 2.3: this is not very crucial but as a suggestion, I would organize it as follows:

2.3.1 Dust deposition samplers

  2.3.1.1 Flat plate sampler

2.3.1.2 Sigma-2 sampler

2.3.1.3 The Modified Wilson and Cooke (MWAC) sampler

2.3.1.4 The Big Spring Number Eight (BSNE) sampler

2.3.2 Free-wing impactor (FWI)

2.3.3 Filter sampler

And I would add the Upward-downward sampler to the Sect 2.3 or 2.3.1 since they are essentially dust samplers that are analysed in the same approach.

Authors' response: The revised manuscript is organized accordingly

Referee's comment- Sect 2.11. Could you add a short description of the inlet biases? From figure 11 it seems that they are negligible? If the differences are so small, is it worth it to show this correction?

The inlet over-/underestimations are corrected for by the formulas given by Belyaev and Levin (1974). The account basically for the fact, that large particles move simply straight into the inlet, while small ones follow the stream lines around. If inlet speed and wind speed differ, this will lead to systematic biases for the large particles with respect to the concentration ratio inside / outside the sampler. Indeed, the effect is not strong here, which is visible in the plots. We show it to demonstrate that this is no major effect here.

Referee's comment- Table S1: I don't understand when it says "Minimum, maximum and median daily basis mass deposition rate". From what I see in the data, it seems that you have reported the mass deposition rate size distribution (average I guess) for each day, but I don't understand where the maximum and minimum are. This happens for other tables in the SI.

Authors' response: Indeed, what is reported in the table is the mass deposition rate size distribution. The referral to the statistical terms was a remainder from a previous version, when not all data was shown. An accordingly correction is made in the revised supplement.

Referee's comment- Fig. 5 and Fig 6. Inconsistent notation in the axis labels

Authors' response: Correction in the revised MS is made.

Referee's comment- Fig. 6. Maybe mention in the caption that the median, percentiles and standard deviations shown there correspond to the variability of the whole campaign for each instrument and bin

Authors' response: The sentence is mentioned in the revised MS.

Referee's comment- Line 464-466 "The deposition rate ratios obtained from the measurements are identical to the deposition velocity ratios, when the sampling time and concentration are the same". What does this mean? Which deposition velocity ratios do you mean in the second case?

In the first part of the sentence, it was referred to the measured deposition rate and their ratios, while on the second part, the (unknown) true deposition velocities were meant. We cannot determine the true value of the deposition velocities, but we can check, whether the models and measurements for an arbitrary pair of samplers operated in parallel work fine by comparing the theoretically and empirically determined velocity ratios. We have modified the manuscripts as follows:

"While without a true reference technique the absolute deposition velocities can't be determined, their ratio between different instruments can be compared theoretically and by measurement. The deposition velocity ratios for a pair of different samplers are identical to the deposition rate ratios obtained from the corresponding measurements (eq. (7)), as long as the sampling time and the aerosol concentration are the same; the latter condition is achieved by the close and parallel sampling."

Referee's comment- Lines 498-492. Table 3 shows that there is a positive dependency in between the concentrations and number deposition rates (r coefficients are positive, so if you increase one variable, the other also does) but it doesn't show a linear correlation since the r2 values are not close to 1 for the first two samplers (particularly the MWAC). However there is a good correlation for the last two samplers. Please clarify this in the text. Also, when it is written "see Figure S 16" it should be "see Figure S 16a"

Authors' response: A clarification has been added in the revised MS. Also, Figure S 16" (now Figure S 22 in the revised MS) changed to "Figure S 16a" (now Figure S 22a) and Figure S 22b is also added in the revised MS.

Referee's comment- As a suggestion, you probably don't need to mention "in the electronic supplement" (e.g: "see Figure S X in the electronic supplement") every time since the label Figure S X already means electronic supplement.

Authors' response: Correction is made in the revised MS.

Referee's comment: - Line 524. "Negligible". Wouldn't it better to use another concept to describe the relation in between the wind speed and the deposition velocities? The analysis shown shows that the wind speed doesn't correlate with the deposition velocity and rate at all but it doesn't necessarily show the effect is negligible.

Authors' response: Correction is made in the revised MS.

Referee's comment- Figure S 9 (and also previous referee comment 32). How can the errors of the campaign average be so high? For the MWAC, as an example, you are basically reporting a range of about 4.5 orders of magnitude for each bin. I suggest checking how errors have been calculated here and reconsidering if it is worth it to call that agreement considering the massive uncertainties.

The term error bar was used to refer to the type of graphical display, but in fact they do not show the error, but the daily variation in terms of the 95 % confidence interval. The daily variation is much higher than the statistical errors (which are shown for example in Figure S10). However, as the wording was misleading we changed it into "The bars show the central 95% confidence interval of the daily variation. The error for each sample size bin is much smaller (e.g., Figure S12 in the revised supplement)."

Referee's comment- Line 555. "Overall, the number concentrations obtained from OPC measurements are slightly higher than the ones". This is not that clear, particularly for the Sigma-2. Please it discuss more.

Authors' response: a more precise sentence stating the behavior of Sigma-2 is added in the revised MS.

Referee's comment - Line 560. Isn't his statement also supported by the Figure S11? If so indicate.

Authors' response: Yes, indeed it is supported by the Figure S11 (Figure S 14 and S 15 in the revised supplement) (and also S 16 and S 17 in the revised supplement). Correction is made in the revised MS.

Referee's comment- Line 564. Legend of the panels a and c and d should be reordered: OPC, Momentum flux, Momentum flux PM10, Wood 1981 and Wood 1981 PM10. This would be make easier to realize that the difference in between applying the correction and not applying it is so small. In addition, I think the axis limits should be chosen differently since more than half of the graph is empty. (5 orders of magnitude are shown while the data only scatters over 2 orders of magnitude in the y axis).

Authors' response: The legends are corrected accordingly in the revised MS. The axis limits are also changed (see in the revised MS).

Referee's comment- Line 569. "Deosition" should be deposition.

Authors' response: Correction is made in the revised MS.

Referee's comment- Fig S11 and S12. Why not to split each of this figures into part a and part b, and having the multi panel with all the information for each sampler under the same title? In the caption of the Fig S11, yellow and green are mentioned while data has been plotted in red and green

Authors' response: The figure is re-arranged according to the comment in the revised. Moreover, correction on caption has been made (see the revised supplement).

Referee's comment- Line 586. "The blue curve shows the concentration measurements by the OPC". This is not necessary since it appears in the legend.

Authors' response: We guess here the referee wanted to refer to 'Line 568' but not to 'Line 586'. Correction is made in the revised MS.

Referee's comment- Figure S 10. Why is there such a massive discrepancy between the Sigma-2 and the rest of the instruments on the 29th of July? Also, the colour choice seems a bit arbitrary, why does cyan appears twice?

Authors' response: We guess here the referee wanted to refer to 'Flat plate', but not to 'Sigma-2'. We agree that on small magnification they were difficult to distinguish, so we replaced the colors.

The mass concentration measured by Flat plate on July 29, 2017 is extremely low when compared with measurements by other samplers. The low value of mass concentration observed with Flat plate for this particular day cannot be explained by other observations, so it has to be considered as an artifact. Therefore, we show the data, but we do not take it into account for further discussion. In addition, we found there is no problem with the color choice (a solid cyan refers Flat plate while a dotted cyan refers to Sigma-2). See also authors' answer to the previous referee's comment (Referee's comment 20: Line 473).

Referee's comment - Line 600. I would mention clearly somewhere that graphs like the Figure 13 but for the other samplers have been presented in the SI, as well as the reason why the BSNE has been shown primarily (see response to referee's pervious comment 48). Also, this section in general needs a bit more of discussion on the agreement or disagreement of the results. For example, the disagreement for the flat plate (I the supplement) is not as significant as the others.

Authors' response: a paragraph is added in the revised MS mentioning size distribution comparison of other samplers. Moreover, the reason why BSNE has been shown primarily is also explained. Also, sentence on the agreement or disagreement of the result is added.

Referee's comment- Line 601-602 "The reasons for this weak correlation – in particular in comparison to the ones 601 from Sigma2 and BSNE – remain unexplained by now". Does this mean the reasons for the disagreement between the different samplers and the OPC/filters? If so explain better.

The manuscript was modified to

"The reasons for this weak correlation between the filter sampler and the OPC measurements – in particular compared to the stronger correlation between Sigma-2 and BSNE with the OPC – are not clear."

Referee's comment- Line 638-641. I would add some lines summarizing the comparisons shown in the electronic supplement.

Authors' response: A paragraph summarizing the comparisons is added in the revised MS

**Other general comments:**

Referee's comment- In some occasions, a few examples of a comparison are shown in the MS, while the bulk of the data is presented in the electronic supplement. (e.g: Fig 11, S11 and S12). I would state it more clearly in the main text for the sake of simplicity.

Authors' response: An explanation is added in the revised MS.

Referee's comment- Many figures, particularly in the electronic supplement, don't have a legend (and it is explained instead in the caption). In general it is better to have legends than explanations in the caption if possible.

Authors' response: Legends are inserted for some of figures in the revised electronic supplement.

Dear Referee #2,

We thank you for the critical comments and suggestions to improve the manuscript (MS). We have considered the comments and modified the MS accordingly. Our detailed responses to the comments are given below.

Referee comment: Although the site is an ideal place to measure dust aerosol in a natural environment, the resuspension of local soil should be considered in comparisons of different sampling techniques. Resuspension of local soil is not externally originated and thus may overestimate natural deposition. Line 414.The description of Figure 5 in 4.1.1 here indicates that the peak is at 16-32 μm. From the figure, however, it is 8-16 μm, please check it.

We agree with the reviewer that in principle there might be local resuspension, which needs to be taken into account when interpreting the results atmosphere-wise. The current paper has a technical focus, and for that, the exact source of the particles is less relevant.

However, we inspected the surrounding of the sampling site in this regard. The site itself is based on a large concrete slab, from which no primary resuspension is expected. In the vicinity (50 m) between the plants, bare sediment is found. However, while taking soil samples, we found that the top 3-5 cm consisted nearly of millimeter- to centimeter-sized pebbles only, with all the fine material removed. Therefore, if there is not exceptional high wind occurring, we would not expect a strong contribution from the local material.

Indeed 16-32 µm was mistake and now correction is made in the revised MS.

[revised manuscript text omitted]